# Discovery and validation of sub-threshold genome-wide association study loci using epigenomic signatures

Xinchen Wang[1,2,3], Nathan R Tucker[4], Gizem Rizki[1], Robert Mills[4],
Peter HL Krijger[5,6], Elzo de Wit[5,6], Vidya Subramanian[1], Eric Bartell[1],
Xinh-Xinh Nguyen[4], Jiangchuan Ye[4], Jordan Leyton-Mange[4], Elena V Dolmatova[4],
Pim van der Harst[7,8], Wouter de Laat[5,6], Patrick T Ellinor[2,4],
Christopher Newton-Cheh[2,4,9], David J Milan[4]*, Manolis Kellis[2,3]*,
Laurie A Boyer[1]*

[1]Department of Biology, Massachusetts Institute of Technology, Cambridge, United States; [2]Broad Institute of MIT and Harvard, Cambridge, United States; [3]Computer Science and Artificial Intelligence Laboratory, Massachusetts Institute of Technology, Cambridge, United States; [4]Cardiovascular Research Center, Massachusetts General Hospital, Boston, United States; [5]Hubrecht Institute-KNAW, University Medical Center Utrecht, Utrecht, Netherlands; [6]University Medical Center Utrecht, Utrecht, Netherlands; [7]Department of Cardiology, University Medical Center Groningen, University of Groningen, Groningen, Netherlands; [8]Department of Genetics, University Medical Center Groningen, University of Groningen, Groningen, Netherlands; [9]Center for Human Genetic Research, Massachusetts General Hospital, Boston, United States

*For correspondence: dmilan@
mgh.harvard.edu (DJM); manoli@
mit.edu (MK); lboyer@mit.edu
(LAB)

Competing interests: The
authors declare that no
competing interests exist.

Reviewing editor: Gilean
McVean, Oxford University,
United Kingdom

**Abstract** Genetic variants identified by genome-wide association studies explain only a modest proportion of heritability, suggesting that meaningful associations lie 'hidden' below current thresholds. Here, we integrate information from association studies with epigenomic maps to demonstrate that enhancers significantly overlap known loci associated with the cardiac QT interval and QRS duration. We apply functional criteria to identify loci associated with QT interval that do not meet genome-wide significance and are missed by existing studies. We demonstrate that these 'sub-threshold' signals represent novel loci, and that epigenomic maps are effective at discriminating true biological signals from noise. We experimentally validate the molecular, gene-regulatory, cellular and organismal phenotypes of these sub-threshold loci, demonstrating that most sub-threshold loci have regulatory consequences and that genetic perturbation of nearby genes causes cardiac phenotypes in mouse. Our work provides a general approach for improving the detection of novel loci associated with complex human traits.

## Introduction

Genome-wide association studies (GWAS) hold the promise of identifying genetic loci that drive complex disease, however realizing this goal has been challenging due to the modest effect sizes of most common variants that require extremely large cohorts to detect with significance. The recent demonstration that disease-associated single nucleotide polymorphisms (SNPs) reside preferentially in enhancer elements provides a unique opportunity to leverage epigenomic maps of regulatory elements for understanding the function of known GWAS loci and for prioritizing new loci missed in

**eLife digest** Most complex traits are governed by a large number of genetic contributors, each playing only a modest effect. This makes it difficult to identify the genetic variants that increase disease risk, hindering the discovery of new drug targets and the development of new therapeutics.

To overcome this limitation in discovery power, the field of human genetics has traditionally sought increasingly large groups, or cohorts, of afflicted and non-afflicted individuals. Studies of large cohorts are a powerful approach for discovering new disease genes, but such groups are often impractical and sometimes impossible to obtain. However, it has become possible to complement the genetic evidence found in disease association studies with biological evidence of the effects of disease-associated genetic variants.

Wang et al. focus specifically on genetic sites, or loci, that do not affect protein sequence but instead affect the non-coding control regions. These are known as enhancer elements, as they can enhance the expression of nearby genes. These loci constitute the majority of disease regions, and thus are extremely important, but their discovery has been hindered by our relatively poor understanding of the human genome.

Chemical modifications known as epigenomic marks are indicative of enhancer regions. By studying the factors that affect heart rhythm, Wang et al. show that specific combinations of epigenomic marks are enriched in known trait-associated regions. This knowledge was then used to prioritize the further investigation of genetic regions that genome-wide association studies had only weakly linked to heart rhythm alterations. Wang et al. directly confirmed that genetic differences in "sub-threshold" regions indeed alter the activity of these regulatory regions in human heart cells. Furthermore, mutating or perturbing the predicted target genes of the sub-threshold enhancers caused heart defects in mouse and zebrafish.

Wang et al. have demonstrated that epigenome maps can help to distinguish which sub-threshold regions from genome-wide association studies are more likely to contribute to a disease. This allows for the discovery of new disease genes with much smaller cohorts than would be needed otherwise, thus speeding up the development of new therapeutics by many years.

current studies (*Ernst et al., 2011*; *Cowper-Sal·lari et al., 2012*; *Maurano et al., 2012*; *Trynka et al., 2013*). Despite increasingly large GWAS cohort sizes, the current catalog of genome-wide significant loci still explains only a modest proportion of the heritability for any given trait, with an excess of low p-value loci still below the genome-wide significance threshold (*Arking et al., 2014*). These observations suggest that many more signals with 'sub-threshold' significance remain to be identified, however, the recognition of biologically relevant sub-threshold loci is hindered by a higher false positive rate (*Hindorff et al., 2009*; *Maher, 2008*; *Altshuler et al., 2008*). Thus, new computational approaches that integrate genetic data with genome-wide epigenomic profiles are needed to use existing cohorts to discover new loci and genes that influence complex traits and diseases.

Here, we use epigenomic maps of 127 tissues from the Roadmap Epigenomics Project as a guide to systematically identify biologically relevant sub-threshold variants (Roadmap Epigenomics Consortium, 2015). As proof of concept, we focused on two cardiac traits with clinical significance: electrocardiographic QT interval reflecting myocardial repolarization and QRS duration reflecting cardiac conduction. These two traits have a clear tissue of origin and published GWASs have reported over a hundred QT/QRS loci, making these traits ideal for testing variants with sub-threshold significance (*Supplementary file 1*) (*Hindorff et al., 2009*; *Maher, 2008*; *Altshuler et al., 2008*). In particular, variation within QT interval length plays an important role in human disease, where extreme QT prolongation is associated with sudden cardiac death and can occur as an unintended side effect of many non-cardiac medications (*Rabkin et al., 1982*; *Heist and Ruskin, 2010*). We combine genome-wide maps of cardiac enhancer activity with the results from a large study of QT interval duration to identify dozens of novel QT loci with sub-threshold statistical significance. We provide multiple lines of evidence to show that these sub-threshold loci can alter enhancer activity, and we implicate specific genes through which these loci act to influence QT interval length. Importantly, we demonstrate

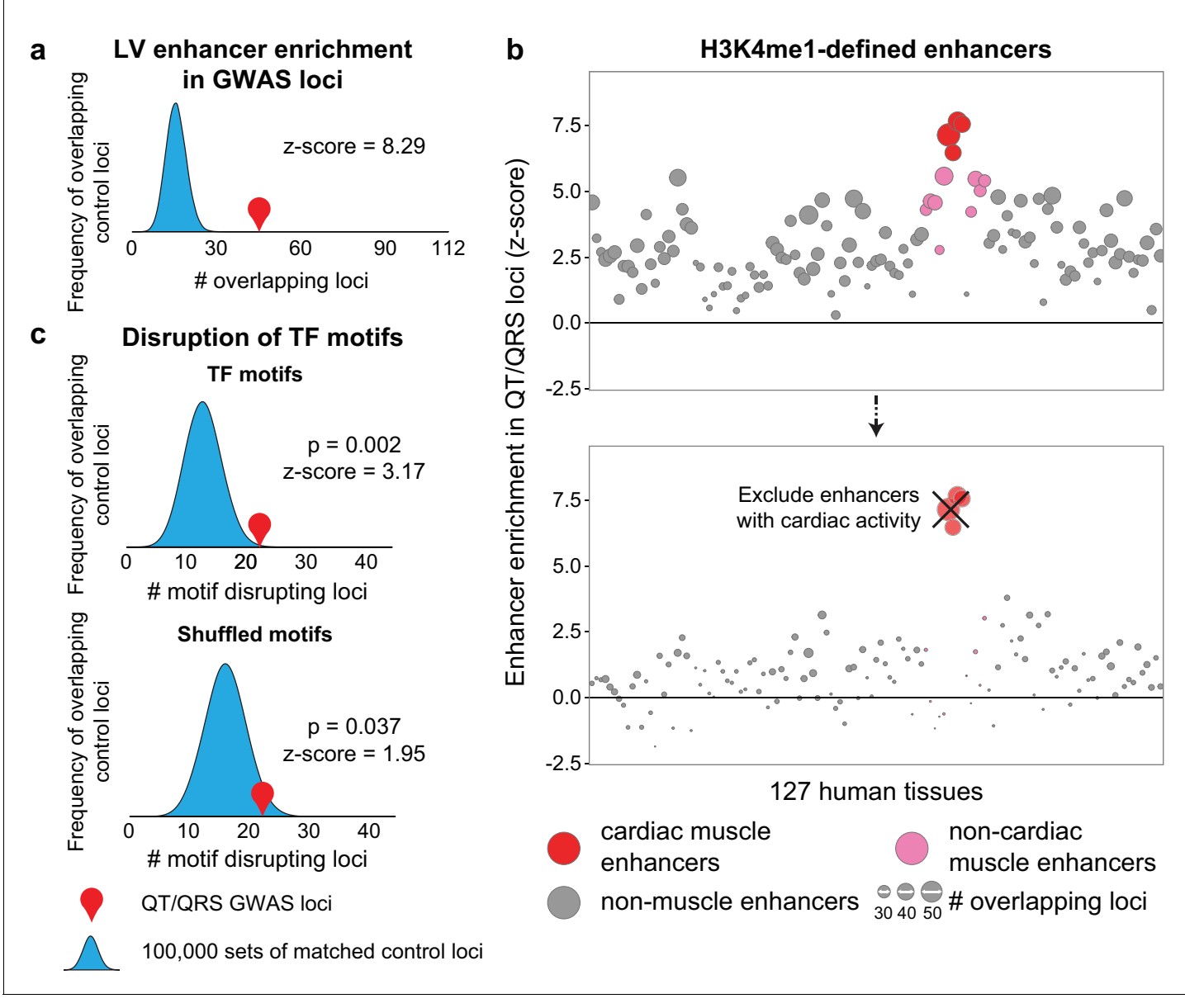

**Figure 1.** GWAS repolarization loci preferentially overlap cardiac enhancers. (a) Enrichment of human left ventricle enhancers in 112 QT/QRS loci. The number of loci that contain a SNP overlapping an enhancer are computed for the 112 QT/QRS loci, and compared against 100,000 permutations of randomly sampled control loci matched for LD block size (number of SNPs), MAF, distance to nearest gene, number of nearby genes, and presence on genotyping array. (b) *Top*, Enrichment of enhancers from 127 human tissues in QT/QRS loci. *Bottom*, Enrichment of enhancers from non-cardiac tissues for QT/QRS loci is substantially weaker following removal of enhancers active in any of the four cardiac tissues. (c) *Top*, QT/QRS SNPs are more likely to disrupt motifs corresponding to expressed TFs compared to 100,000 sets of matched control loci. *Bottom*, Weaker enrichment was observed between repolarization and matched control loci when the sequence of the TF motif was randomly shuffled and re-mapped to the genome (10,000 permutations).

The following figure supplements are available for figure 1:

**Figure supplement 1.** 112 QT/QRS loci overlap enhancers more significantly than other genomic regions in adult left ventricle.

**Figure supplement 2.** QT/QRS loci overlap enhancers more significantly than other genomic regions in non-LV cardiac tissue.

**Figure supplement 3.** Enrichment of cardiac enhancers remains high in the subset of 53 genome-wide significant ($p<5\times10^{-8}$) QT/QRS loci.

that epigenetic signals can distinguish true biological signals from noise, thus bypassing the higher false positive rate that has previously hindered study of sub-threshold loci. We expect our work will uncover new genes involved in cardiac electrophysiology, aid in the identification of patients at risk for sudden cardiac death, and enable development of new treatments for susceptible individuals. More broadly, our work demonstrates the power of integrating epigenomics with existing GWAS to discover sub-threshold genetic loci and novel genes associated with complex human disease.

## Results

### QT/QRS-associated variants are enriched in cardiac enhancers

We compiled a list of 112 QT/QRS loci from the NHGRI GWAS database (accessed July 2013, *Supplementary file 1*) and identified SNPs in strong linkage disequilibrium ($r^2$>0.8) using genotype data from the 1000 Genomes Project (Phase 1, CEU population) (*1000 Genomes Project Consortium, 2010*). We also collected GWAS loci from a later meta-analysis of QT interval studies, published in June 2014 by Arking et al., which we held out from the aforementioned 112 QT/QRS loci as a validation dataset for subsequent analyses (*Arking et al., 2014*). Because only 22 of 112 loci (20%) harbor SNPs that overlap exons, we examined whether QT/QRS variants are enriched in predicted enhancer elements across the genome using chromatin maps across 127 tissues generated by the Roadmap Epigenomics Project including adult left ventricle (LV), adult right ventricle (RV), fetal heart (FH) and adult right atrium (RA) (*Roadmap Epigenomics Consortium, 2015*). QT/QRS variants have greatest overlap with predicted enhancers (as defined by high levels of H3K4me1 and low H3K4me3 using ChromHMM) from the four cardiac tissues compared to the other 123 non-cardiac tissues (red circles, *Figure 1b*, *Supplementary file 1*) (*Ernst et al., 2011*). This enrichment persists over randomly sampled sets of control loci with matched genetic properties such as minor allele frequency, number of SNPs in LD, distance to nearest gene, number of nearby genes, and presence on an Affymetrix 660W genotyping array (*Figure 1a*, Materials and methods). Enhancers from the LV showed the strongest enrichment of the four cardiac tissues (z-scores=7.67, empirical p<$1\times10^{-5}$, $10^5$ permutations), demonstrating that an unbiased analysis can resolve the causal tissue with high precision, as QT interval and QRS duration are primarily reflective of myocardial repolarization in the ventricles.

Focusing on the left ventricle, we analyzed the enrichment of both coding annotations using GENCODE and non-coding annotations using individual chromatin marks and chromatin states defined by ChromHMM as well as DNase I hypersensitivity (DHS) maps available in heart tissue (*Ernst et al., 2011*; *Harrow et al., 2012*; *Thurman et al., 2012*). We observed that intergenic enhancers are the most strongly enriched annotated genomic region (z-score > 7.5) in QT/QRS loci, followed by gene transcription regions (z-score between 3 and 6) (*Figure 1—figure supplements 1* and *2*). This enrichment increased significantly (z-score from 7.67 to 9.31 for left ventricle) when restricting the analysis to 'strong' enhancers (i.e. H3K4me1 enhancers that are also marked by H3K27ac). Our results indicate that predicted enhancers are highly informative for annotating trait-associated variants compared to other classes of genomic regions.

We next asked whether LV enhancers that overlap QT/QRS loci have features that distinguish them from putative LV enhancers identified by ChromHMM that do not overlap QT/QRS loci (*Figure 2*). First, we considered the density of H3K27ac marks, as the co-enrichment of H3K4me1 and H3K27ac correlates with strong enhancer activity (*Creyghton et al., 2010*; *Rada-Iglesias et al., 2011*). We found that the 65 enhancers overlapping 45 QT/QRS loci have a 3.1-fold higher H3K27ac density compared to non-GWAS LV enhancers (p=$1.53\times10^{-4}$). In fact, incorporating H3K27ac into ChromHMM enhancer predictions resulted in substantially greater enrichment of QT/QRS loci (z-score = 9.31 vs. 7.67 for left ventricle); 44 of the 45 QT/QRS loci overlap an H3K27ac-defined 'strong' enhancer. QT/QRS LV enhancers are also more likely to be marked by either H3K4me1 or H3K27ac in at least one of the other three heart tissues (fetal, right atrium, right ventricle) compared to non-GWAS LV enhancers (p-values between 0.004 and 0.04, *Figure 2*) and less likely to be active in non-cardiac tissues (p=$9\times10^{-3}$, *Figure 2*).

Left ventricular QT/QRS enhancers are significantly more hypomethylated than predicted LV enhancers not overlapping QT/QRS loci (hypomethylation p=$1.07\times10^{-6}$, hypermethylation p=0.60, *Figure 2*). Similar to H3K27ac, CpG hypomethylation correlates with increased enhancer activity, possibly through modulation of TF binding site accessibility (*Hon et al., 2013*; *Stadler et al., 2011*).

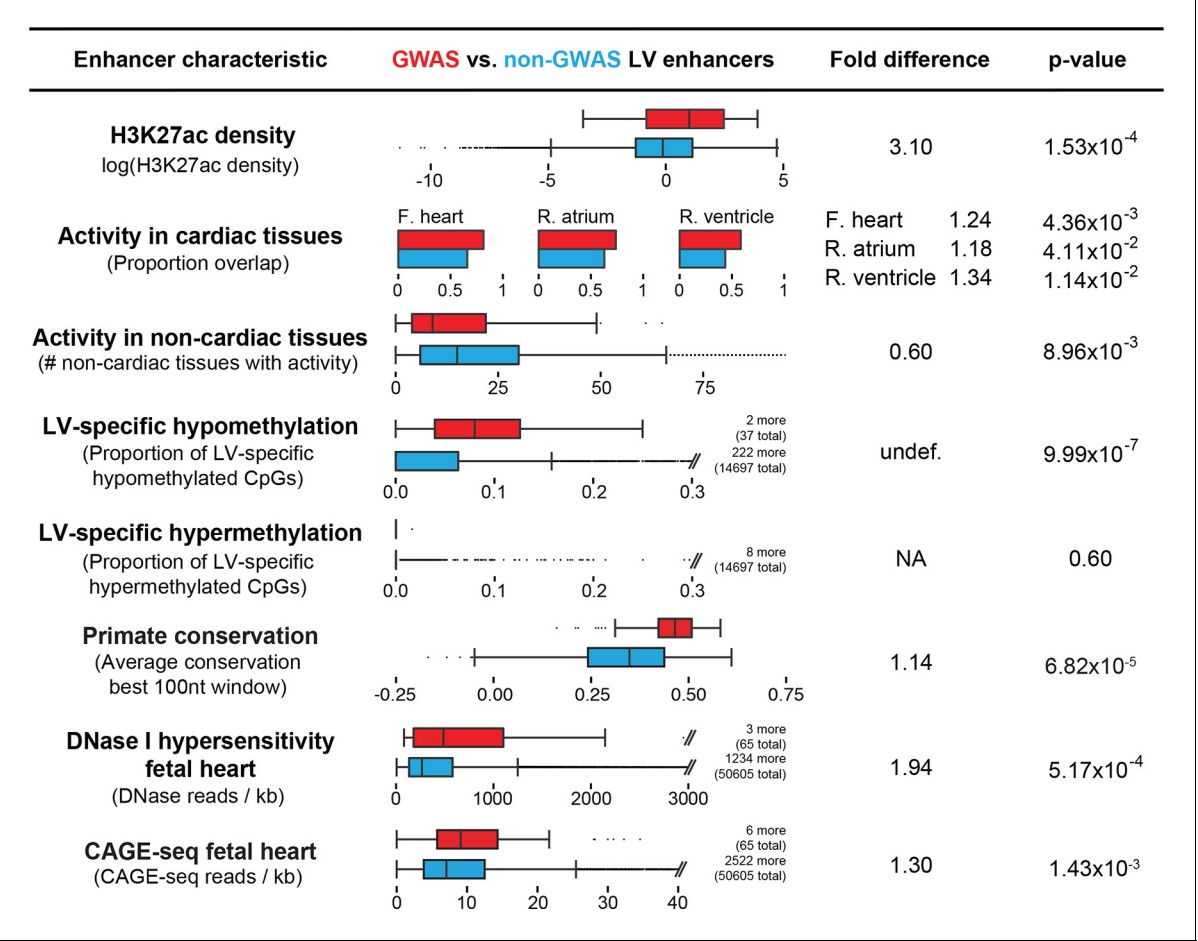

**Figure 2.** Enhancers overlapping QT/QRS loci differ in functional characteristics from all enhancers. Several functional characteristics were compared between enhancers overlapping QT/QRS loci (red) and non-GWAS left ventricle enhancers (blue). Fold change represents fold change between median values for the two groups, and p-values were calculated using the Mann-Whitney U test. See Materials and methods for comparison methodology between GWAS QT/QRS enhancers and non-GWAS enhancers for each functional or epigenomic feature. For primate conservation, LV enhancers (blue) were size-matched (+/-1 kb) to GWAS enhancers to control for skewed enrichments driven by larger GWAS enhancer size.

Consistent with this idea, 22 of the 45 GWAS loci contain an enhancer SNP that alters a predicted motif for a cardiac-expressed TF (empirical p=0.002, $10^5$ permutations) (*Figure 1c*). Moreover, QT/ QRS GWAS enhancers are enriched for DHS and Cap Analysis Gene Expression (CAGE) signals in human fetal heart, both of which are marks of greater enhancer activity (*Figure 2*) (*Thurman et al., 2012*; *Andersson et al., 2014*). Finally, QT/QRS left ventricular enhancers show significant evolutionary conservation across the primate lineage compared to non-GWAS LV enhancers (p=6.82x$10^{-5}$ compared to $10^5$ size-matched sets of LV enhancers), suggesting that perturbation of these enhancers is under stronger negative selection. Taken together, QT/QRS loci preferentially overlap conserved enhancers that show cardiac-restricted activity, suggesting that common variants associated with these loci play roles in regulating cardiac functions that drive human phenotypes.

## Common features in GWAS cardiac enhancers identify novel sub-threshold loci

Current GWAS loci collectively explain only a small fraction of the estimated heritability of a complex trait in part due to strict Bonferroni thresholds for multiple hypothesis testing (p<5x$10^{-8}$) and the limited statistical power of existing studies to discover variants with modest effect sizes (*Maher, 2008*; *Yang et al., 2011*). We hypothesized that knowledge of the genomic properties associated with existing GWAS loci can guide the search for additional genetic signals that cannot be detected

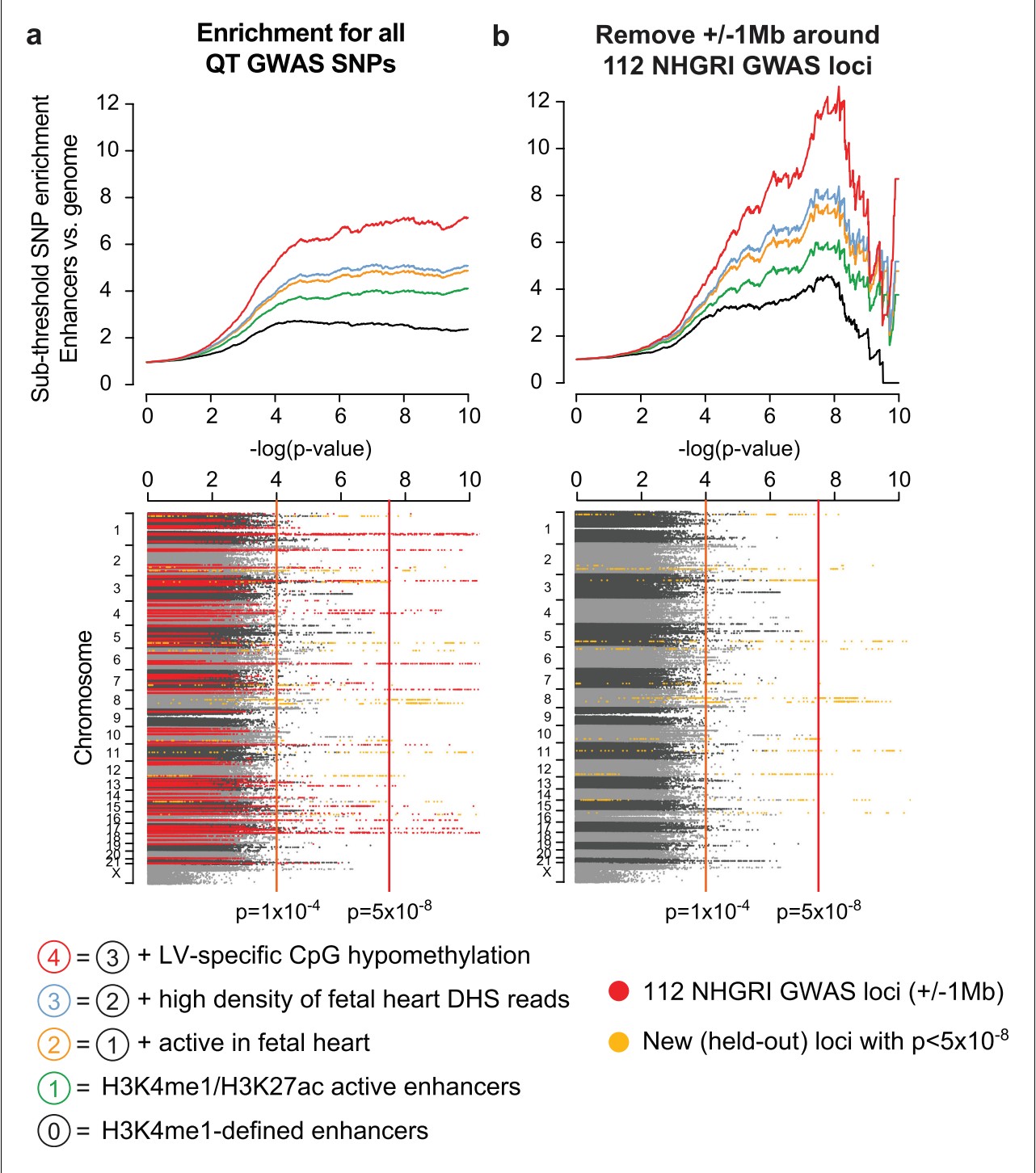

**Figure 3.** Cardiac enhancers harbor additional sub-threshold QT loci. (**a**) *Top*, Enhancer characteristics learned on above-threshold QT/QRS loci from *Figure 2* are predictive for additional sub-threshold loci (colored lines). Each point on a curve represents the fold difference in proportion of SNPs with p-value below the cutoff in the enhancer set versus the whole genome. *Bottom*, Manhattan plot of p-values for all SNPs from ***Arking et al. (2014)*** QT interval GWAS. 112 QT/QRS loci and all SNPs within 1 Mb are highlighted in red. Genome-wide significant loci newly discovered by Arking et al. and not in the 112 QT/QRS loci are highlighted in yellow. (**b**) *Top*, Enrichment signals for sub-threshold SNPs in left ventricle enhancers persists following removal of the 112 GWAS loci and nearby SNPs (+/- 1 Mb). *Bottom*, Manhattan plot of p-values for all SNPs from ***Arking et al. (2014)*** QT interval GWAS following removal of 112 QT/QRS loci and all SNPs within 1 Mb. Genome-wide significant loci newly discovered by Arking et al. and not in the 112 QT/QRS loci are highlighted in yellow.

*Figure 3 continued on next page*

*Figure 3 continued*

The following figure supplements are available for figure 3:

**Figure supplement 1.** High density of fetal heart DNase I hypersensitivity reads in LV enhancers is robustly informative for identifying enriched sets of sub-threshold loci.

**Figure supplement 2.** Enrichment in the sub-threshold significance range can be observed using only SNPs nearby known above-threshold loci.

**Figure supplement 3.** Sub-threshold loci associated with QT interval length are enriched in H3K4me1-defined left ventricle enhancers.

**Figure supplement 4.** Enhancers harbor additional sub-threshold loci associated with Alzheimer's disease and LDL cholesterol.

without increasing GWAS cohort sizes, and that these loci with weaker 'sub-threshold' p-values (i.e. $0.05 > p > 5 \times 10^{-8}$) might reveal novel genes and biological pathways that contribute to complex disease. To test this idea, we used SNP summary statistics from the *Arking et al. (2014)* QT interval GWAS study we had earlier held out as a validation dataset (*Arking et al., 2014*). These summary statistics include the 112 QT/QRS loci identified by prior GWASs (red dots, *bottom*, *Figure 3*), as well as loci that reach genome-wide significance in the larger meta-analysis cohort but were not discovered in any previous GWAS (and therefore were not included in the 112 QT/QRS loci used for enrichment analyses above, gold dots, *bottom*, *Figure 3*). We observed that active LV enhancers are strongly enriched for loci harboring SNPs with p-values between $1 \times 10^{-4}$ and $5 \times 10^{-8}$ (*Figure 3a*, black line). Furthermore, the combination of functional features identified for above-threshold QT/QRS enhancers (*Figure 2*) substantially improves sub-threshold locus enrichment across a wide range of p-value thresholds (*Figure 3a*, colored lines, *Figure 3—figure supplement 1*).

Whether the enrichment of SNPs in the sub-threshold significance range represents linkage disequilibrium with existing above-threshold GWAS SNPs or novel biologically relevant loci remains an unresolved question (*Maurano et al., 2012*). In fact, an enrichment analysis using only SNPs nearby above-threshold GWAS loci produced a strong enrichment signature in the sub-threshold significance range (*Figure 3—figure supplement 2*). To distinguish between the two possibilities, we took a conservative approach and removed all SNPs within 1Mb of the initial 112 QT/QRS loci. Remarkably, the enrichment for LV enhancers persists and increases in the sub-threshold range (i.e. $p = 1 \times 10^{-4}$ to $5 \times 10^{-8}$, *Figure 3b*), likely due to removal of nominally significant SNPs that are in LD with above-threshold QT/QRS loci and do not represent true association signals. The enrichment for active LV enhancers in sub-threshold loci is not driven by biases in MAF, LD block size, distance to nearest gene, number of nearby genes, or presence on a SNP genotyping array (*Figure 3—figure supplement 3*). In total, we identified 2075 SNPs with $p < 1 \times 10^{-4}$ that are independent of the 112 published QT/QRS loci, of which 208 SNPs overlap LV enhancers (*Supplementary file 2*).

## Epigenomic prioritization identifies sub-threshold loci with molecular functions

Because the enrichment of sub-threshold SNPs in cardiac enhancers suggests that epigenetic prioritization can be used as a starting point for more in-depth investigations of sub-threshold signals from GWAS, we sought to directly test the molecular hypothesis that these sub-threshold loci impact the transcriptional regulation of cardiac genes (*Figure 4a*). We grouped all 2075 sub-threshold SNPs using linkage disequilibrium data (minimum $r^2 = 0.2$) to identify 287 independent sub-threshold loci in the genome (Materials and methods). We prioritized loci where a sub-threshold SNP overlapped an active LV enhancer and either (i) also overlapped a fetal heart DNase I hypersensitivity peak or (ii) was an expression quantitative trait locus (eQTL) for a nearby gene. In total, we cloned allele-specific enhancer fragments from 22 cardiac enhancers that overlap SNPs from 18 independent sub-threshold loci, and performed quantitative luciferase assays in human iPSC-derived cardiomyocytes to determine whether the sub-threshold SNP genotypes influence enhancer activity (Materials and methods). We observed that 13 of 18 sub-threshold loci (72.2%) contain an enhancer that drives luciferase activity in an allele-specific manner (*Figure 4b,d*, *Figure 4—figure supplement 1*). Moreover, we estimate that between 51.1%-89.8% (95% Bayesian confidence interval) of prioritized sub-threshold loci show allele-specific activity on transcription, suggesting that the majority of

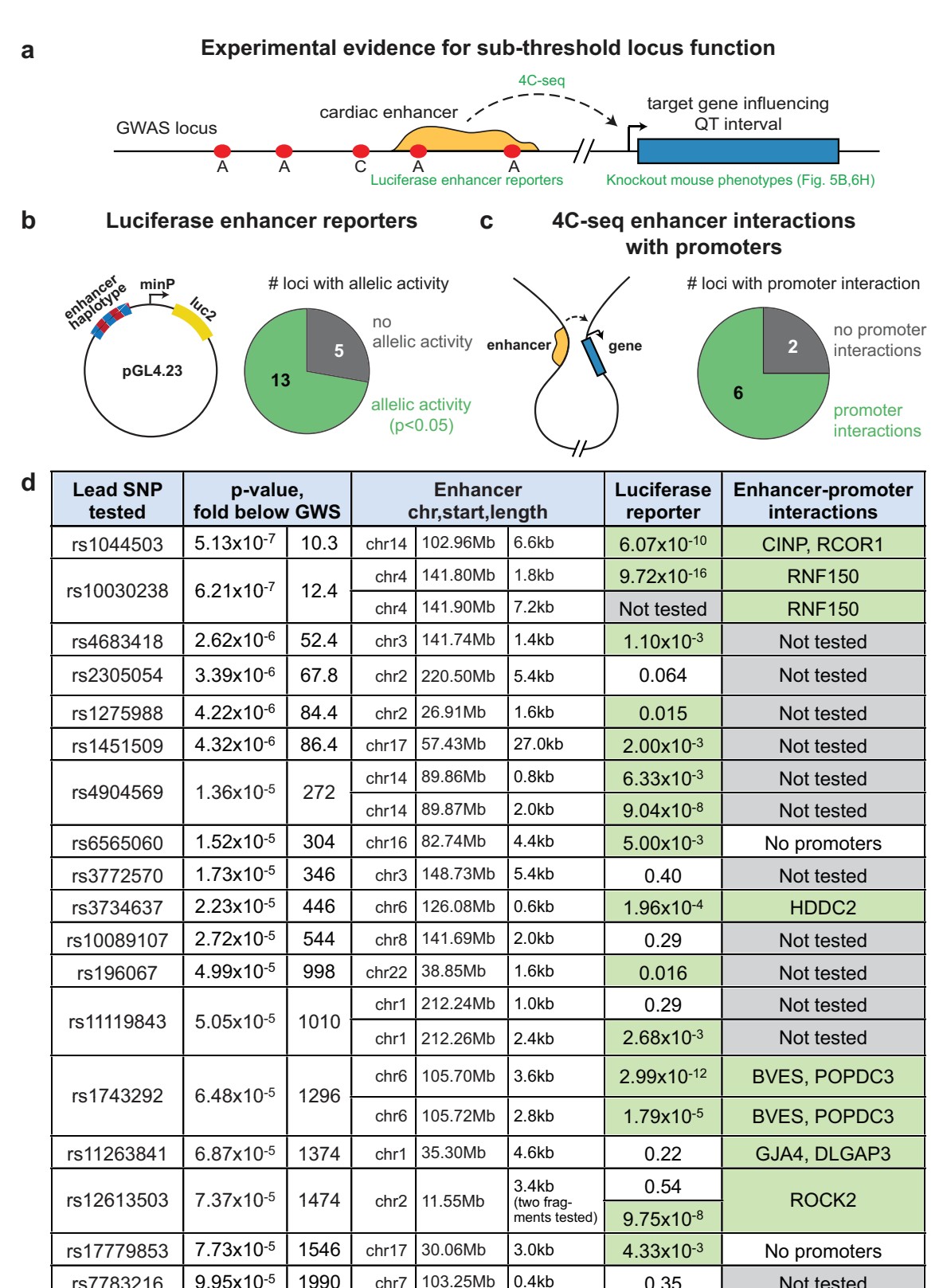

**Figure 4.** Sub-threshold loci prioritized by epigenomics alter enhancer activity. (**a**) Model detailing how sub-threshold SNPs overlapping enhancers can affect QT interval. *Green text: methods used to test mechanistic step in model.* (**b**) Summary of luciferase enhancer reporter experiments. *Left,*

*Figure 4 continued*

luciferase enhancer reporter construct. *Right,* number of loci tested in panel d that exhibits significant allelic activity (p<0.05 between two haplotypes). (c), *Left,* schematic of a 3-D enhancer-promoter chromatin interaction detectable by 4C-seq. Right, number of loci tested in panel d where an enhancer-promoter interaction is observed in human iPS-derived cardiomyocytes by 4C-seq. (d) Experimental evidence that sub-threshold SNPs alter enhancer activity and that sub-threshold enhancers interact with gene promoters. *Fold below GWS column* represents degree to which sub-threshold locus is below genome-wide significance (5x10$^{-8}$); *Luciferase reporter column* colored green if significant allelic difference in activity (p<0.05, **Figure 4—figure supplement 1**); *Enhancer-promoter interactions column* colored green if there is a detectable enhancer-promoter interaction by 4C-seq (**Figure 4— figure supplement 3**).

The following figure supplements are available for figure 4:

**Figure supplement 1.** Sub-threshold SNP alleles affect enhancer activity.

**Figure supplement 2.** Luciferase enhancer reporter assay for sub-threshold SNPs outside enhancers.

**Figure supplement 3.** 4C-seq interactions with 10 enhancers in 8 sub-threshold loci.

---

sub-threshold loci identified by epigenomic prioritization do in fact have an impact on transcriptional enhancer activity.

We also performed chromosome conformation capture combined with high-throughput sequencing (4C-seq) to experimentally test whether predicted enhancers in sub-threshold loci can form contacts with promoters, and to identify potential target genes of sub-threshold enhancers. We used 4C-seq to test ten predicted enhancers from eight sub-threshold loci in human iPSC-derived cardiomyocytes (*van de Werken et al., 2012*). Eight enhancers in six loci formed enhancer-promoter interactions in the proximal 500 kb region (**Figure 4c**, **Figure 4 — figure supplement 3**). These analyses provides evidence that the novel QT loci enhancers have regulatory activity and that the sub-threshold SNPs identified in our analyses can alter the activity of cardiac enhancers.

## Epigenomic prioritization discriminates biologically relevant sub-threshold loci

We next tested whether epigenomic prioritization can distinguish biologically relevant sub-threshold loci by comparing properties of sub-threshold loci that do or do not overlap cardiac enhancers. From the 287 independent sub-threshold loci in the genome, we selected two subsets to compare: 60 loci that contain sub-threshold SNPs directly overlapping predicted active LV enhancers, and as a negative control, 129 sub-threshold loci that do not contain any SNPs (r$^2$>0.2) overlapping a cardiac enhancer.

### Evidence from genome-wide association studies

We reasoned that if sub-threshold loci that overlap active cardiac enhancers represent true biological signals, they should have stronger GWAS association signals than the negative control set. We present multiple lines of evidence supporting this hypothesis below (**Figure 5a**):

1. The 60 enhancer-overlapping sub-threshold loci have significantly stronger p-values than the 129 negative control loci, despite the application of the same p=1x10$^{-4}$ threshold for both sets (p=1.95x10$^{-5}$, *left*, **Figure 5a**).
2. 9 of the 60 enhancer-overlapping sub-threshold loci are among the loci that reach genome-wide significance in the larger held out meta-analysis cohort (and not included in the 112 QT/ QRS loci used for enrichment analyses in **Figure 1** and **2**), compared to only 3 of the 129 sub-threshold loci that do not overlap enhancers (6.45-fold enrichment, p=1.92x10$^{-3}$, *middle*, **a**).
3. The 60 enhancer-overlapping sub-threshold loci are more likely to reach nominal significance (p<0.05) in a related GWAS study of QRS duration (see Materials and methods for individuals shared between both studies)(van der Harst, unpublished). In the QRS duration GWAS, p-values are available for 56 of 60 enhancer-overlapping sub-threshold QT loci and 110 of 129 negative control sub-threshold loci. 31 of 56 (55.4%) enhancer-overlapping sub-threshold loci are nominally significant in the QRS GWAS, a rate 2.9-fold higher than the 129 negative control loci (21 of 110 loci, p=3.28x10$^{-6}$, *right*, **Figure 5a**), suggesting that epigenetic prioritization is more likely to identify sub-threshold SNPs that replicate in subsequent GWASs.

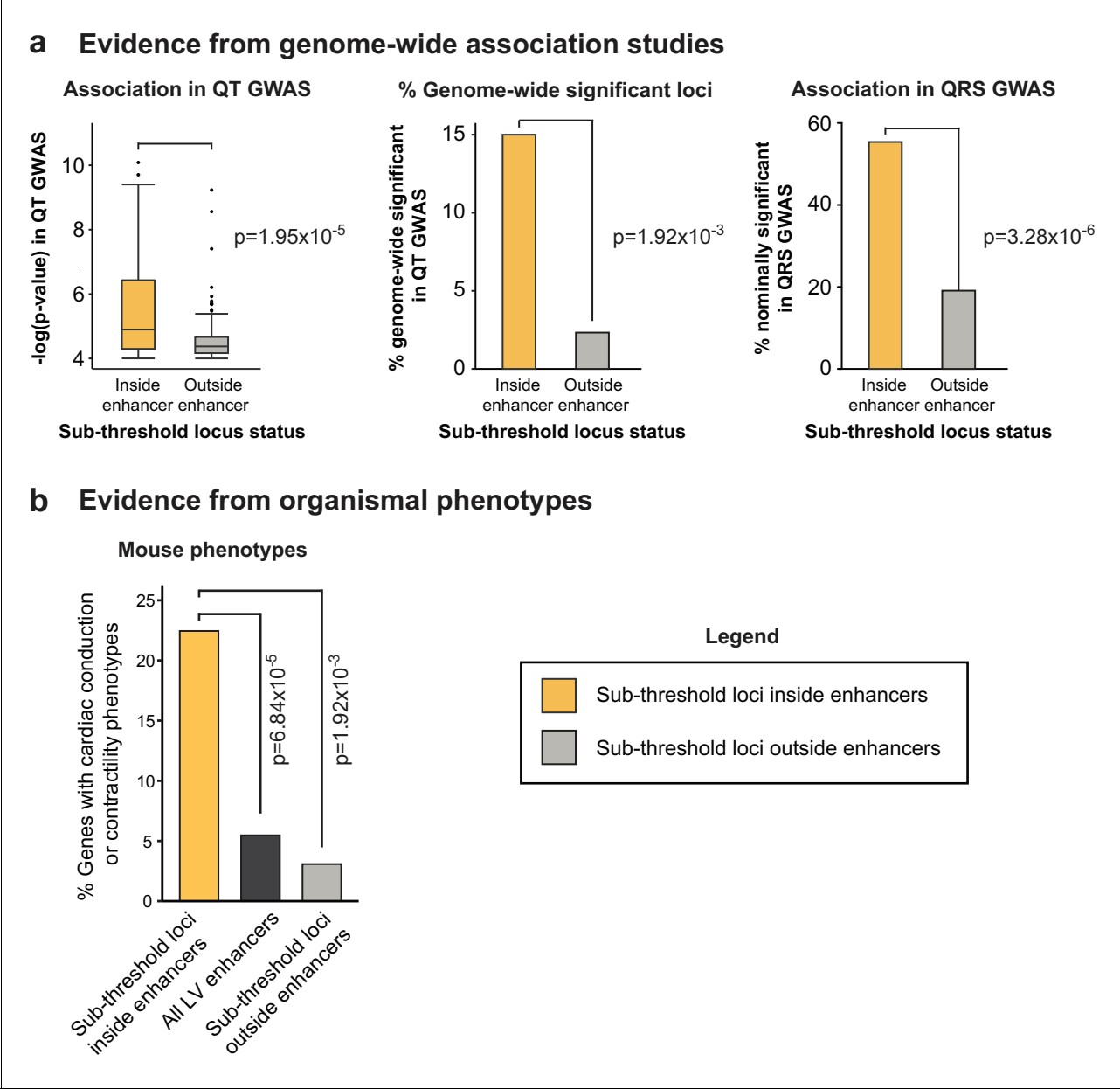

**Figure 5.** Epigenomic prioritization distinguishes biologically relevant sub-threshold loci (**a**) *Left,* Sub-threshold loci overlapping enhancers have significantly stronger association signals than loci outside enhancers in the QT interval GWAS. *Middle,* Loci overlapping enhancers have significantly more likely to be newly genome-wide significant in the held-out QT interval GWAS, than loci outside enhancers. *Right,* Sub-threshold loci overlapping enhancers are significantly more likely to be nominally significant (p<0.05) in QRS GWAS than sub-threshold loci not overlapping enhancers. (**b**) Genetic perturbation of genes with predicted links to 60 enhancer-overlapping sub-threshold loci are significantly more likely to result cardiac conduction or contractility phenotypes than genes linked to all LV enhancers and genes nearby non-enhancer overlapping sub-threshold loci.

These analyses demonstrate that genome-wide maps of predicted enhancers can facilitate the detection of true sub-threshold loci.

## Evidence from organismal phenotypes

Our identification of a high-confidence set of sub-threshold loci based on epigenomic signals provides a unique opportunity to discover new genes that contribute to cardiac electrophysiological

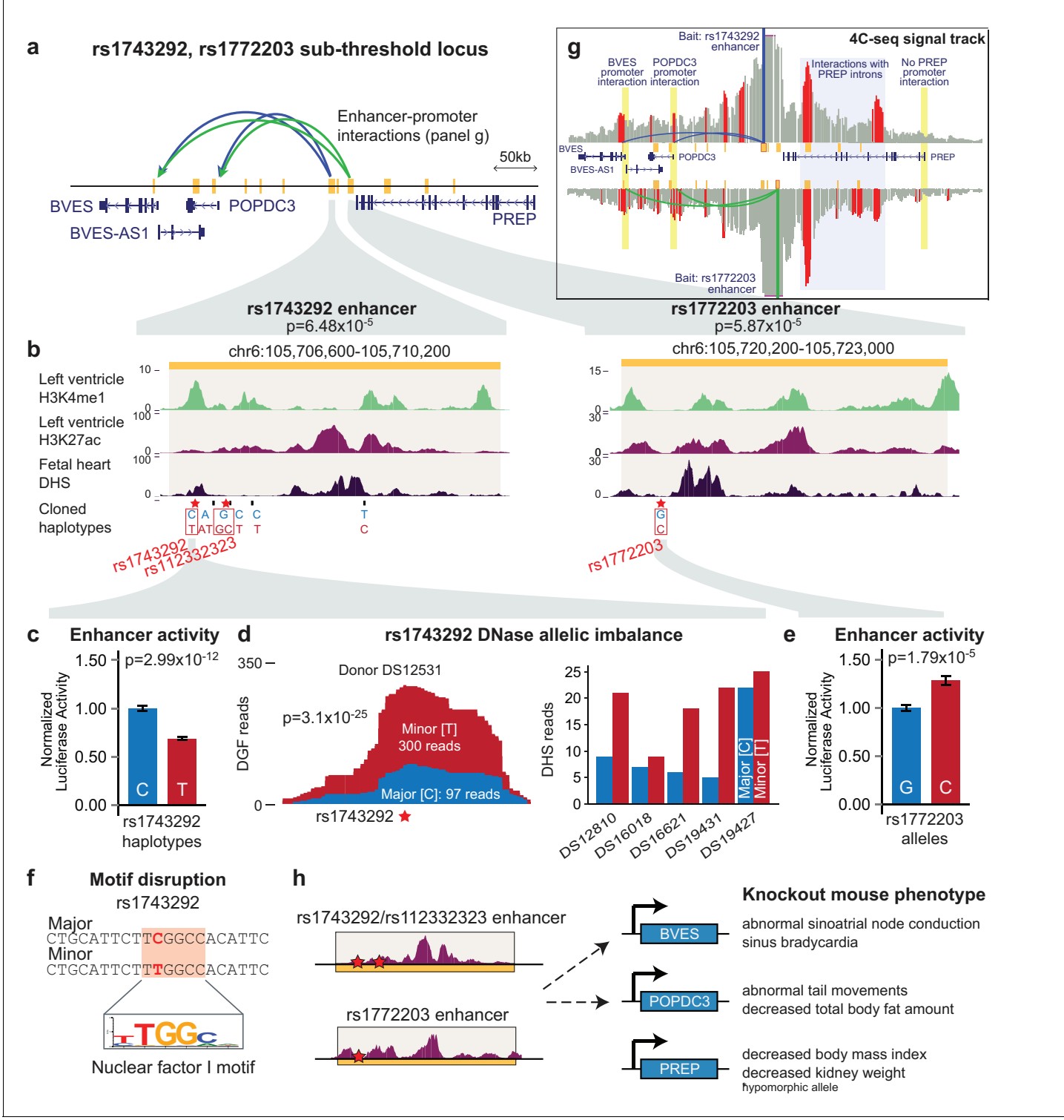

**Figure 6.** The rs1743292/rs1772203 sub-threshold locus disrupts activity of cardiac enhancers that interact with BVES, a gene important for cardiac electrophysiology. (**a**) Overview of rs1743292/rs1772203 sub-threshold locus. Gold rectangles represent predicted active LV enhancers, blue and green lines represent enhancer promoter interactions from the rs1743292 and rs1772203 enhancers, respectively (see panel g). (**b**) Detailed view of cardiac enhancers overlapping rs1743292 (left) and rs1772203 (right). (**c**) rs1743292 haplotypes differing at 6 SNPs (listed at bottom of panel b) affect activity of cardiac enhancer in human iPSC-derived cardiomyocytes, n=24 per haplotype. Error bars represent standard error of the mean. (**d**), *Left,* rs1743292 alters level of DNase I hypersensitivity in a heterozygous human fetal heart sample. *Right,* Allelic imbalance of DHS reads at rs1743292 observed for 5 of 5 human individuals. (**e**) rs1772203 allele affects activity of cardiac enhancer in human iPSC-derived cardiomyocytes, n=16 per allele. Error bars

*Figure 6 continued on next page*

*Figure 6 continued*

represent standard error of the mean. (**f**) rs1743292 SNP overlaps a predicted nuclear factor I (NF-I) motif. (**g**) 4C-seq analysis of the rs1743292 (blue) and rs1772203 (green) enhancers identifies enhancer-promoter interactions with nearby BVES, BVES-AS1 and POPDC3 genes, and additional enhancer-enhancer interactions within introns in PREP. (**h**) Genetic perturbation of *Bves,* but not *Popdc3* or *Prep* leads to cardiac electrophysiological defects in mouse models.

The following figure supplements are available for figure 6:

**Figure supplement 1.** Expression patterns of BVES, POPDC3 and PREP across 59 human tissues.

**Figure supplement 2.** Knockdown of *bves* in zebrafish leads to ventricular repolarization defects.

traits. As enhancers can regulate genes up to 1Mb away, it is difficult to identify targets using a simple nearest gene approach (*Fullwood et al., 2009*). To circumvent this limitation, we developed a computational enhancer-gene linking method that prioritizes gene targets based on correlated activity patterns between enhancer-gene pairs across 59 human tissues (Materials and methods). Using this approach, we identified 106 candidate genes predicted to be regulated by the 60 enhancer-overlapping sub-threshold loci (*Supplementary file 2*). Notably, 11 of the 15 observed 4C-seq interactions were predicted by our computational approach, compared to 3 of 15 by the commonly applied approach of assigning the enhancer target to the nearest gene.

We used the output of the enhancer-gene linking method to test whether these candidate genes have roles in QT interval. To this end, we studied mouse phenotypes for directed knockouts and genetic perturbations of the 106 predicted gene targets of sub-threshold enhancers. We identified 49 of the 106 genes where mouse mutant models were available with documented phenotypes (*Bello et al., 2015*). Genetic perturbation in 11 of the 49 genes resulted in altered cardiac conduction or cardiac contractility: both processes that are also influenced by genes nearby above-threshold QT interval loci and genes implicated in the Mendelian Long QT syndrome. This represents a 4.11-fold enrichment compared to genes linked to all active LV enhancers (181 of 3311, $p=6.84 \times 10^5$, black bar, *Figure 5b*). In contrast, phenotypes arising from genetic perturbation of LV-expressed genes nearby the 129 negative control sub-threshold loci outside enhancers are 7.30-fold less likely to result in altered cardiac conduction or contractility compared to our 60 prioritized sub-threshold loci ($p=1.92 \times 10^{-3}$, perturbation of 2 of 65 genes nearby the negative control subset have relevant cardiac phenotypes, light grey bar, *Figure 5b*).

The study of biologically relevant sub-threshold loci has been hampered by a high false positive rate that makes the detailed investigation of any sub-threshold locus experimentally more difficult and less attractive than above-threshold loci. The data presented here provide multiple independent lines of evidence that epigenomic signatures can be used to prioritize sub-threshold GWAS loci with a significantly greater likelihood of being biologically relevant.

## Sub-threshold locus at rs1743292/rs1772203 functionally disrupts enhancer activity

Only a very small number of above-threshold GWAS loci, including *SORT1* for LDL cholesterol, the *FTO/IRX3* locus for obesity, and the *SCN5A/SCN10A* locus for QRS duration, have been investigated in detail (*Musunuru et al., 2010*; *van den Boogaard et al., 2012*; *van den Boogaard et al., 2014*; *Arnolds et al., 2012*; *Smemo et al., 2014*). These studies all identified SNPs within non-coding regulatory elements that disrupt expression of a nearby gene that plays a critical role in controlling a human phenotype. In contrast, no sub-threshold locus has been experimentally studied or validated to date. We selected one locus on chromosome 6 where our results from *Figures 4* and *5* suggest that sub-threshold SNPs disrupt enhancer activity and therefore expression of a gene involved in cardiac electrophysiology. We set out to investigate whether this locus can serve as an example for future investigations of other sub-threshold loci.

The sub-threshold locus on chromosome 6 contains 8 SNPs with reported p-values less than $1 \times 10^{-4}$ and another 2 SNPs in LD that do not have calculated p-values. We focused on the 3 SNPs in this locus that overlap active LV enhancers: rs1743292 ($p=6.48 \times 10^{-5}$) and rs112332323 (p-value not available)

that both overlap a 3.6 kb predicted enhancer, and rs1772203 (p=5.87x10$^{-5}$) that overlaps a 2.8 kb predicted enhancer (*Figure 6a,b*). We cloned fragments corresponding to both enhancers upstream of a minimal promoter driving the luciferase gene, and compared luciferase activity between constructs carrying either the major or minor haplotypes at each site (rs1743292 enhancer: *Figure 6c*, rs1772203 enhancer: *Figure 6e*, SNPs differing between cloned constructs listed at bottom of *Figure 6b*). We observed that the activity of both enhancers is dependent on the sub-threshold haplotype: at the rs1743292 enhancer, the major haplotype has 45% greater activity (p=2.99x10$^{-12}$), while the minor haplotype is 28% more active in the rs1772203 enhancer (p=1.79x10$^{-5}$).

In the fetal human heart, rs1743292 overlaps a strong DNase I hypersensitivity peak marking a local region of open chromatin signifying potential transcription factor binding (*DHS track*, *Figure 6b*) (*Neph et al., 2012*). Thus, to provide evidence that the rs1743292 locus alters enhancer activity in humans, we re-aligned the DHS sequencing reads from heterozygous human individuals in an allele-specific manner to assess the difference in the number of reads that map to either allele (*Maurano et al., 2012*). In fetal heart tissue from one individual sequenced to high depth, rs1743292 shows a significant allelic imbalance for DHS reads with 97 reads mapping to the major C allele and 300 reads mapping to the minor T allele (*left, Figure 6d*, p=3.1x10$^{-25}$, binomial test). This trend is consistent in all five additional human individuals heterozygous at rs1743292 sequenced at lower depth (*right, Figure 6d*), suggesting that rs1743292 can affect enhancer activity potentially through altering chromatin accessibility or transcription factor binding. Moreover, using motif analysis, we observed that rs1743292 alters a predicted binding site for the cardiac-expressed nuclear factor NF-I family (*Figure 6f*), which contains a family member (NF-1a) that itself has been associated by GWAS with cardiac electrophysiology (*Ritchie et al., 2013*).

We used 4C-seq to identify genes that could be regulated by the rs1743292 or rs1772203 enhancers. We observed that both enhancers form interactions with promoters of the upstream popeye-domain containing (POPDC) family members *BVES/POPDC1* and *POPDC3*, and with predicted enhancers situated within introns of the downstream *PREP* gene (*Figure 6a,g*). This suggests that both enhancers may contribute to regulating the gene expression of *BVES* and *POPDC3*, of note because the POPDC protein family of transmembrane proteins has recently reported roles in cardiac pacemaking (*Froese et al., 2012*; *Kirchmaier et al., 2012*).

We sought to investigate the roles of the three candidate target genes (*BVES, POPDC3, PREP*) of the rs1743292/rs1772203 locus in regulating myocardial repolarization. Consistent with the genetic association between this locus and QT interval length, we found that mice homozygous for loss-of-function copies of *BVES* exhibit cardiac conduction and pacemaker defects (*Figure 6h*) (*Bello et al., 2015*; *Froese et al., 2012*). In contrast, *POPDC3* and *PREP* mouse loss-of-function models have no reported cardiac abnormalities, and instead show altered body fat, suggesting that this genetic locus alters QT interval length through the *BVES* gene (*Bello et al., 2015*).

Strengthening our evidence implicating *BVES* in QT interval, we observed that across 59 human tissues, *BVES* is most highly expressed in human left ventricle, whereas *POPDC3* has much lower expression in cardiac tissue than skeletal muscle, and *PREP* is constitutively expressed across a wide range of tissues (*Figure 6—figure supplement 1*). We also used antisense morpholino oligonucleotides to knockdown transcripts from the *BVES, POPDC3* and *PREP* orthologs in zebrafish, observing that *bves* knockdown leads to a reproducible shortening of the zebrafish ventricular action potential duration (APD), the cellular correlate of the QT interval, (p=0.002 and 0.09 for two independent morpholino sequences), whereas there is no reproducible difference in ventricular APD following loss of *popdc3* or *prep* transcripts (*Figure 6—figure supplement 2*). Collectively, these data from multiple organisms provide evidence that SNPs within the rs1743292/rs1772203 locus alter QT interval duration through disruption of *BVES* expression.

These results provide evidence that cardiac enhancers can be used to identify novel sub-threshold loci and genes associated with cardiac traits. As demonstrated with the luciferase enhancer reporter assays, and specifically the rs1743292/rs1772203 locus, sub-threshold loci harbor SNPs that affect enhancer activity and regulate genes involved in QT interval. In the current QT interval GWAS, rs1743292 had an effective sample size of 68,900 individuals with 12.76% power to detect the locus at genome-wide significance. To detect rs1743292 at genome-wide significance with 80% power would require a GWAS cohort of 146,700 individuals. Thus, our study demonstrates that genome-wide enhancer maps are a powerful tool for identifying sub-threshold loci with *bona fide* roles in

human cardiovascular physiology that would have remained otherwise unrecognized from existing GWAS cohorts.

## Discussion

A major limitation in the human genetics field is the inability to ascribe function to the vast majority of non-coding SNPs associated with complex human traits. Using enhancer annotations from hundreds of cell types and tissues, we find ~50% of QT/QRS GWAS loci overlap enhancers, and that these enhancers share common characteristics, including H3K27ac marks, CpG hypomethylation, and greater evolutionary conservation. The high density of common variation we observed in non-coding enhancers may be due to weaker evolutionary selection against the subtle phenotypes that arise from disruption of transcriptional regulatory units compared to the more severe disruption of protein-coding sequences commonly observed in rare Mendelian diseases.

Studies of genetic heritability have indicated that many additional loci lie below the genome-wide significance threshold (*Yang et al., 2011*). Our study contributes fundamental insights to overcoming the difficult problem of discovering the biologically relevant sub-threshold genetic signals that are orders of magnitude weaker than discovered by traditional GWAS. Three prior studies have observed the general enrichment of either sub-threshold SNPs or SNPs that explain a disproportionately high amount of heritability in cell type-specific regulatory elements (*Maurano et al., 2012*; *Finucane et al., 2015*; *Gusev et al., 2014*). However, our study is unique in demonstrating the advantage of combining different epigenomic features to produce greater enrichments of sub-threshold loci. Critically, no previous study to our knowledge has implicated any specific sub-threshold locus in any complex human trait, whereas we establish that 13 of the 18 sub-threshold loci tested in this study are capable of altering enhancer activity. We also leverage GWAS summary statistics and genetic perturbations in mouse to demonstrate that epigenetic marks can discriminate true positive sub-threshold signals from noise, a key problem that, until now, has prevented the study of these loci. Finally, we perform an in-depth molecular dissection of the rs1743292/rs1772203 sub-threshold locus and implicate the popeye-domain containing family of transmembrane proteins in regulating myocardial repolarization. The study of above-threshold GWAS loci is generating more biological insights on new causal genes contributing to human disease, however there remains a wealth of untapped signals in the sub-threshold region. The work presented here represents a first step towards deciphering this signal and opens the door for the discovery of greater numbers of disease loci, genes, and pathways.

Our study focused on QT interval and QRS duration due to their clear tissue of origin and a wealth of existing GWAS data, however we believe our approach could generalize to any well-powered GWAS on any trait. To this end, we chose two recently published, well-powered GWASs that relate to human diseases affecting large segments of the population: LDL cholesterol levels and Alzheimer's disease. For both traits, we observed the enrichment of SNPs well into the sub-threshold significance range, that the enrichment signature persists following removal of all above-threshold loci, and that functional features that improve enrichment of QT-associated sub-threshold loci are also effective when applied to sub-threshold loci associated with LDL cholesterol and Alzheimer's disease (*Figure 3—figure supplement 4*). These results suggest that epigenomics can be applied more broadly to identify new loci with sub-threshold statistical significance from GWAS of many complex human diseases. One important future extension of this work would be to build a formal machine learning classifier that can be first trained on above-threshold GWAS loci before being applied to quantitatively rank sub-threshold loci by predicted biological relevance.

Finally, investigating the differences between above-threshold and sub-threshold loci to elucidate the factors that drive loci to different degrees of association with a trait will be an important area of future investigation. Many reported genome-wide significant loci have been discovered by GWAS despite low power, likely due to the existence of many other variants of similar effect that go undetected, termed the 'winner's curse', and thus this difference could be driven in part by random chance. However, we also hypothesize that sub-threshold loci with weaker effect sizes may act in different pathways from loci with stronger effect sizes, and that sub-threshold variants could have weaker effects on gene expression.

In summary, our results provide a critical roadmap for the systematic analysis and re-analysis of genome-wide association studies to prioritize novel biologically relevant loci with weak association

signals. As demonstrated with the rs1743292/rs1772203 locus, these loci would otherwise require substantially greater cohort sizes to reach statistical significance. Thus, we expect that this approach can be exploited to broadly improve the understanding of the biological pathways that contribute to complex human traits and disease.

# Materials and methods

## Identifying GWAS loci associated with cardiac traits

We compiled a list of all SNPs associated with electrocardiographic QT interval (reflecting myocardial repolarization) or QRS duration (reflecting cardiac conduction) from the NHGRI GWAS catalog of published GWAS (accessed on July 09, 2013), and removed loci identified from studies with small sample sizes (<5000 individuals). As the GWAS catalog reports SNPs with $p<1\times10^{-6}$, we performed a sensitivity analysis using only loci with $p<5\times10^{-8}$ to demonstrate that two different cut-offs does not meaningfully affect enrichment results for left ventricle (*Figure 1—figure supplement 3*). We used genotype data from the 1000 Genomes project to identify all SNPs in LD ($r^2>0.8$, CEU population) with the lead SNPs. For cases where two lead SNPs were in LD with each other (i.e. different studies reported different SNPs from the same haplotype block), we merged the resulting loci. To avoid over-counting, if the sets of LD SNPs from two independent lead SNPs overlapped, we randomly assigned each of the shared LD SNPs to only one of the two lead SNPs.

## RNA-seq data and enhancer annotations

### Epigenome roadmap datasets.

Processed RNA-seq data for 59 human tissues and enhancer annotations (for 127 H3K4me1-defined and 88 'strong' H3K4me1/H3K27ac-defined enhancer sets) were downloaded from the Roadmap Epigenomics Project (*Roadmap Epigenomics Consortium, 2015*). Initial analyses across all 127 tissues were performed on cardiac enhancers defined by ChromHMM by the Roadmap Epigenomics Project using five chromatin modifications including H3K4me1 but not H3K27ac (15-state model). 'Strong' cardiac enhancers, available for a subset of 88 tissues, were defined by ChromHMM by the Roadmap Epigenomics Project using six chromatin modifications including both H3K4me1 and H3K27ac (18-state model).

### Human differentiated cardiomyocyte RNA-seq dataset.

hESCs were differentiated to cardiomyocytes as previously described (*Elliott et al., 2011*) and were obtained from David Elliott at Monash University. RNA was extracted using TRIzol reagent according to the manufacturer's instructions. 10 µg RNA was used for library construction according to Illumina RNA-seq library kit with minor modifications. Briefly, mRNA was isolated using Dynabeads mRNA Purification Kit (Invitrogen, Catalog #61006) followed by fragmentation and ethanol precipitation. First and second strand synthesis were performed followed by end repair, A-tailing, paired end adaptor ligation and size selection on a Beckman Coulter SPRI TE nucleic acid extractor. 200-400 bp dsDNA was enriched by 15 cycles of PCR with Phusion High-Fidelity DNA Polymerase (NEB, Catalog #M0530) followed by gel purification of 250 bp fragments from the amplified material. Amplified libraries were sequenced on an Illumina GAIIx sequencer. Reads were mapped against the hg19 version of the human genome using RSEM v. 1.2.3 and bowtie v. 0.12.7 using flags "rsem-calculate-expression –phred64-quals -p 4 –output-genome-bam –calc-ci –paired-end –bowtie-chunkmbs 1024, without in-silico polyA addition to the transcripts.

## Enrichment of genomic features in QT/QRS loci

We used genomic features annotated by combinations of histone modifications (e.g. enhancers and promoters using ChromHMM by the Roadmap Epigenomics Project) or by GENCODE (e.g. protein-coding exons). Previous studies have compared the number of GWAS SNPs overlapping a feature against the number expected for a randomly chosen region of similar size (*Maurano et al., 2012*; *Hnisz et al., 2013*). However, this approach does not control for biases associated with the location of GWAS SNPs. We controlled for these biases by following the Variant Set Enrichment approach where we generate a background distribution for genomic feature enrichment in loci around sets of 112 randomly sampled control lead SNPs (*Cowper-Sal·lari et al., 2012*). We chose control lead

SNPs from a genome-wide genotyping array (Affymetrix 660W) matched for size of the LD block (+/- 5 SNPs), minor allele frequency of the lead SNP (+/- 0.1), distance to the nearest gene (+/- 25 kb if outside gene), and number of nearby genes within a +/- 500 kb interval (+/- 3 genes). We also considered differences in local GC content (+/-25 nt) but did not observe a strong difference between GWAS and control lead SNPs (p=0.06). To calculate enrichment of genomic regions in GWAS loci, we compared the number of GWAS loci that overlapped an enhancer to 100,000 sets of equally sized randomly sampled control lead SNPs. The 112 GWAS SNPs compiled from the NHGRI GWAS catalog includes 57 loci with p-values between $1\times10^{-6}$ and $5\times10^{-8}$ that have a higher false positive rate. In a sensitivity analysis, we examined the subset of 55 loci that met the more stringent $5\times10^{-8}$ statistical threshold and found that sets of cardiac enhancers (specifically fetal heart and adult left ventricle) were also most highly enriched in these loci compared to the 123 non-cardiac tissues (*Figure 1—figure supplement 3*).

## Comparing differences between QT/QRS-associated LV enhancers and all LV enhancers

### H3K27ac, DNase I Hypersensitivity and CAGE-seq read enrichment

To score the presence of epigenomic marks in enhancers, we averaged the wig signal tracks over every enhancer with the UCSC bigWigAverageOverBed tool. Fold difference in signals between QT/QRS enhancers and all LV enhancers were calculated by comparing the median signal values of the two groups. P-values were calculated using the Mann-Whitney U test. *Activity in other cardiac and non-cardiac tissues:* Overlap with enhancers in other tissues was calculated using the intersectBed function in the BEDTools suite (*Quinlan and Hall, 2010*). *CpG hypomethylation and hypermethylation:* Whole-genome bisulfite sequencing data for 37 human tissues, including the left ventricle, was obtained from the Roadmap Epigenomics Project (*Roadmap Epigenomics Consortium, 2015*). We identified LV-specific hypo and hypermethylated CpGs as those that differed in percent methylation with the mean of 34 non-cardiac tissues by both (i) 2 standard deviations and (ii) at least a difference in absolute percent methylation of 15 percent. *Evolutionary Conservation:* We calculated evolutionary conservation of enhancers using the methodology outlined by *Nord et al. (2013)*(*Nord et al., 2013*). Briefly, we first identified the 100 bp region of each enhancer with greatest average evolutionary conservation across primates (primate subset of 46-way phyloP conservation track obtained from UCSC). To quantify differences in evolutionary conservation of GWAS enhancers against all LV enhancers, we randomly selected 1000 size-matched sets of LV enhancers (size within +/-1 kb of corresponding QT/QRS enhancer), as the 100 bp segment of greatest conservation in longer enhancers is statistically more likely to have greater conservation than a shorter segment.

## Comparing differences in TF motif disruption

We obtained TF motif instances in the human genome (hg19) for 651 human motifs from the ENCODE project (*ENCODE Project Consortium, 2012*), and filtered these to only consider 287 motifs that correspond to TFs expressed in the left ventricle (>1 RPKM by RNA-seq). We quantified the number of QT/QRS loci containing a SNP that disrupted an enhancer motif corresponding to an expressed TF in the left ventricle, and compared this against randomly sampled sets of control loci matched for MAF, LD block size, distance to the nearest gene and presence on the Affymetrix 660 W genotyping array.

## Enrichment of QT SNPs below genome-wide significance in enhancers

### Enrichment analysis

We used a sliding -log(p-value) threshold from 0 to 10 with steps of 0.1. At each cut-off, we computed the proportion of SNPs in enhancers with p-values more significant than the cut-off (foreground) against the proportion of SNPs in the whole genome. *Grouping SNPs in LD.* For each pair of SNPs, if the two SNPs are in LD ($r^2>0.2$, CEU population from 1000 Genomes project) we remove the SNP with the weaker p-value.

## Enrichment of LDL cholesterol and Alzheimer's disease-associated sub-threshold loci in enhancers

Summary GWAS data for LDL cholesterol was obtained from *Willer et al. (2013)*, and summary GWAS data for Alzheimer's disease (AD) was obtained from *Lambert et al. (2013)*(*Lambert et al., 2013*). Enrichment analyses were performed as described above for QT interval. For enrichment of Alzheimer's disease-associated SNPs, the region encompassing the HLA locus was excluded (chr6:24,182,924–34,537,546 in hg19), as this region contained approximately 25% of all low p-value SNPs ($p < 1 \times 10^{-5}$) in the genome therefore and could skew enrichment results.

The liver tissue was chosen for LDL cholesterol enrichment based on biological relevance. Tissue choice for AD SNPs was made using genome-wide enrichment analyses performed by *Gjoneska et al. (2015)*. For this analysis, we chose the second-most enriched tissue from Gjoneska et al. (peripheral blood monocytes, with most significant p-value) instead of the most enriched tissue (peripheral blood mononuclear cells, PBMCs, with second-most significant p-value) because the enrichment of AD SNPs in PBMC enhancers was substantially weaker than peripheral blood mono-cytes following removal of SNPs within the HLA locus. For AD GWAS, removal of SNPs within +/- 1 Mb of above-threshold loci was performed using 13 above-threshold loci with $p < 5 \times 10^{-8}$ (Stage 1 analysis) listed in Table 2 of Lambert et al. For LDL cholesterol analyses, we first attempted to remove all SNPs within +/- 1 Mb of above-threshold loci reported in *Supplementary files 2 & 3* of Willer et al., however many SNPs with $p < 5 \times 10^{-8}$ remained. Therefore, we performed LD pruning ($r^2 > 0.2$ from CEU population) on summary-level p-value data from Willer et al. to define above-threshold loci and then removed 68 unique genomic intervals from the analysis. Enhancer functional characteristics applied to the enhancer sets were chosen based on the availability of additional data for the chosen tissue. DNase I hypersensitivity data not available for human liver, and genome-wide CpG methylation data was not available for peripheral blood monocytes.

## Comparison of QT sub-threshold loci in QRS GWAS data

To assess whether QT sub-threshold loci overlapping enhancers are more likely to represent true biological signals, we queried the p-values of these loci in a related GWAS of QRS duration. In total, the QT GWAS we used to identify the sub-threshold loci consisted of 76,061 individuals, while the QRS GWAS queried consisted of 60,255 individuals. We compared the total sizes of each cohort used in the two studies and calculated that a minimum total of 46,452 individuals must be different between the two studies. Specifically, there are at least 31,129 individuals present in QT GWAS that are not present in the QRS GWAS, and at least 15,323 individuals present in the QRS GWAS that are not present in the QT GWAS.

We used summary-level p-value data from the QRS GWAS testing four clinically applied QRS traits: Sokolow-Lyon, Cornell, 12-lead-voltage duration products, and QRS duration.

For each SNP, the assigned p-value represented the minimum p-value across these four traits. For each sub-threshold locus, we identified all SNPs in strong LD ($r^2 > 0.8$, CEU population from 1000 Genomes project), and assigned the p-value as the minimum of all p-values for LD SNPs in the QRS GWAS data.

## Identifying candidate genes near sub-threshold loci using activity correlation across human tissues

From the Roadmap Epigenomics Project, we were able to obtain matching 'strong' enhancer anno-tations and RNA-seq data for 59 of the 127 tissues, including LV. For each LV enhancer, we consid-ered all genes with expression $\geq 1$ RPKM in LV and *in vitro* differentiated human cardiomyocytes and distance within +/-500 kb as potential targets. We then split the RNA-seq data for the 59 tissues into two groups, depending on whether the enhancer is present or absent in each tissue, and applied a one-sided Mann-Whitney U test to ask whether each potential target gene showed signifi-cantly greater expression in tissues where the enhancer was active. Genes differentially expressed between tissues with active and inactive enhancers ($p < 0.05$) were considered computationally-deter-mined potential target genes. For determining targets of sub-threshold enhancers, we first filtered our set of sub-threshold enhancers to remove those unlikely to be associated with QT interval. To do this, we excluded sub-threshold SNPs if the -log(p-value) was lower than 80% of the -log(p-value) of the most statistically significant SNP in LD ($r^2 > 0.2$), as these are unlikely to be causal.

## Cardiac phenotypes for genes with mutations in mouse

For sub-threshold loci overlapping enhancers, and the set of all active LV enhancers, we identified nearby genes using the enhancer-gene linking method described above. This methodology was not applicable to the 129 sub-threshold loci that do not overlap enhancers, and therefore we identified the two nearest genes within 1 Mb using GREAT v2.0.2 and selected only genes with expression in adult human left ventricle data (>1 RPKM). Mouse orthologs of human genes were identified using the Ensembl Genes 79 database through BioMart, and all queries of the MGI mouse phenotypes database were made between April 26, 2015 and May 6, 2015. We used three search terms relevant to QT interval: 'ventricle muscle contractility', 'cardiac contractility' and 'conduction' (excluding non-cardiac conduction terms).

## Quantifying allelic imbalance at SNPs

We used DNase I hypersensitivity and digital genomic footprinting data from the ENCODE and Roadmap Epigenomics Projects because samples were sequenced to a greater depth than the chromatin modification ChIP-seq data, and there were data available from more individuals *Roadmap Epigenomics Consortium, 2015*. To quantify allelic imbalance, we mapped DHS/DGF reads to a version of the human genome (hg19) downloaded from the UCSC genome browser with all SNPs (dbSNP141) masked by ambiguous nucleotides (N's) using Bowtie2 (v2.2.0, flags: -N 1, –sensitive, –end-to-end, –no-unal). As genotypes were not available, we considered a sample heterozygous at a particular SNP if reads from the hg19-defined reference and alternate alleles each mapped to 3 or more unique positions. Using this methodology, we observed the median difference in reads mapping to the reference versus alternate alleles to be 0. In total, reads mapped to the reference allele more often than alternate at 6537 of 13,3826 heterozygous SNPs, and vice versa at 5884 of 13,826 heterozygous SNPs, with equal numbers of reads mapping to both alleles at the remaining 1405 SNPs. To quantify statistical significance of allelic imbalance at SNPs, we followed *Maurano et al. (2012)* and considered only SNPs with more than 21 reads. We performed a binomial test under the null hypothesis where reads map to both alleles at equal frequency, followed by Benjamini-Hochberg multiple testing correction across all heterozygous enhancer-overlapping SNPs.

## 4C-Seq Methods

Human iPSC-derived cardiomyocytes (iCMs) (Cellular Dynamics, Catalog #CMC-100-010-001) were thawed according to manufacturer's instructions and diluted to a final plating density of $0.2 \times 10^6$ cells per mL with plating medium (Cellular Dynamics, Catalog#CMM-100-110-001). After 7 days in culture, iCMs were homogenized using a douncer, cross-linked and further processed as 4C template using DpnII as the first restriction enzyme and Csp6I as the second enzyme following the procedure outlined in *van de Werken et al. (2012)*. The median spacing between GATC fragments (recognized by DpnII) in the hg19 human genome is 264 nt. Sequencing of the 4C-Seq library was performed on an Illumina HiSeq 2000, and sequencing reads were aligned to a reduced genome consisting of sequences that flank DpnII restriction sites. Primer sequences used for sequencing the 4C-seq library are listed in *Supplementary file 3*. The human genome (hg19) was used as reference genome for mapping 4C sequence captures. Non-unique sequences that flank a restriction site were removed from the analysis.

To map 4C-seq reads to the genome, we first binned reads according to the reading primers used in each lane. We allow a single mismatch in the reading primer that overlaps the primary restriction cut site (DpnII). The binned sequences were mapped to an in silico library of potential fragment ends generated based on the restriction enzymes used for the 4C template preparation. We did not allow any mismatch in the fragment-end, and for analysis we focused on the unique fragends only (excluding repetitive fragment ends). As biases from sequencing yield or restriction cutting may be introduced by 4C-seq, we computed 4C-seq coverage in a genomic region by averaging mapped reads in running windows of 21 4C-seq fragment-ends. For peak-calling in a single 4C experiment, we perform explicit background modeling of the up- and downstream genomic regions independently. We assume that in a completely unstructured chromatin fiber the contact probability monotonically decreases as a function of the distance to the viewpoint. We model this by performing monotonic regression of the 4C signal as a function of the distance to

the viewpoint. For this we use the R package isotone, which implements the monotonic regression (*Mair et al., 2009*). We then compare the observed 4C signal to the predicted value from the background model and call the extremes that reach a significance threshold as peaks. For a given threshold q and a distribution F of residuals from the background model, every observation greater than Q3(F)+q*IQR(F), where Q3 is the third quartile of F and IRF(F) the inter-quartile range, is considered significant.

## Generating enhancer reporter constructs

Sub-threshold loci were considered candidates for testing by the luciferase reporter assay if the sub-threshold SNP overlapping the active LV enhancer either (i) overlaps a fetal heart DNase I hypersensitivity site, or (ii) is an eQTL in the left ventricle (i.e. the SNP genotype is associated with differential expression of a nearby gene). We generated allele-specific enhancer constructs using two strategies outlined below: (i) PCR from genotyped heterozygous individuals, or (ii) direct synthesis of enhancer fragments. (i) *Enhancer cloning from heterozygous individuals:* We designed primer sequences to clone the entire predicted enhancer sequence defined by ChromHMM, and appended a 5'CACC sequence to forward primers to permit directional TOPO cloning. We designed primer sequences to clone fragments of up to 3 kb. For enhancers annotated as larger than 3 kb, we either selected a 3 kb fragment centered at the region of greatest histone modification density (H3K4me1, H3K27ac), or generated multiple fragments spanning the enhancer. Primer sequences and samples for human genomic DNA (Coriell Cell Repositories) are listed in *Supplementary file 3*. We PCR amplified enhancers from human genomic DNA using Q5 High-Fidelity DNA Polymerase (NEB, Catalog #M0491S) and purified fragments corresponding to the correct length using a QIAquick Gel Extraction Kit (Qiagen, Catalog #28706). (ii) *Direct synthesis of enhancer fragments:* Enhancer fragments up to 1 kb in size were chosen so that the fragment covers both the sub-threshold SNP as well as peak within the DNase I hypersensitivity signal, and a 5'CACC sequence was appended to permit directional TOPO cloning. Fragments were synthesized using the gBlocks Gene Fragments service from Integrated DNA Technologies (sequences are listed in *Supplementary file 3*). Enhancer fragments from both methods were cloned into Gateway-compatible entry vectors using a pENTR/D-TOPO Cloning Kit (Life Technologies, Catalog # K2400) and transformed into TOP10 *E. coli* bacteria following manufacturers guidelines. We used Sanger sequencing to verify that purified entry vectors carried enhancers with the correct insertion orientation and no mutations beyond the expected polymorphisms. Entry vectors were then Gateway-cloned using LR Clonase II Plus (Life Technologies, Catalog # 12538-120) into a Gateway-converted pGL4.23 destination vector (Promega, Catalog # E8411) for luciferase assays in human cell lines (*Fisher et al., 2006*). We used Sanger sequencing to confirm a second time the correct enhancer orientation and sequence inside the destination vectors.

## Human cardiomyocyte luciferase assays

Human iCMs (Cellular Dynamics, Catalog #: CMC-100-010-001) were thawed according to manufacturer's instructions and diluted to a final plating density of $0.2 \times 10^6$ cells per mL with plating medium (Cellular Dynamics, Catalog#: CMM-100-110-001). 96-well tissue culture treated plates were coated with 0.1 mL of 0.1%(w/v) gelatin per well and incubated at 37°C for at least two hours. The gelatin solution was aspirated off and wells rinsed with 100 uL of PBS, aspirated, and let sit in the tissue culture hood. Using a multichannel pipette, 100 uL of cells were seeded per well to obtain a target density of $20 \times 10^3$ iCMs. The plates were kept on a flat bench at room temperature for 10-15 minutes to allow for cells to settle down uniformly, followed by incubation at a tissue culture incubator set at 37°C and 7% $CO_2$. 48 hours post-seeding, the iCM plating medium was replaced with 100 uL of Maintenance Medium (Cellular Dynamics, Catalog #:CMM-100-120-001). The Maintenance Medium was replaced every other day.

3-4 days post-plating, iCMs began beating spontaneously and 7 days post-plating, they formed electrically connected syncytial layers that beat simultaneously. At this stage, the cells were transfected with the appropriate Luciferase reporter constructs and controls for downstream analyses. Media was replaced an hour before transfections. For each well, 95 ng of enhancer firefly Luciferase reporter (cloned into pGL4.23, Promega) and 5 ng of Renilla Luciferase transfection control vector (pGL4.73, Promega) was mixed with 10 ul of Opti-MEM (Life Technologies, Catalog #:51985-034).

0.2 uL of Viafect transfection reagent was added to the DNA/Opti-MEM mixture. After mixing, the transfection cocktail was incubated at room temperature for 5 min and 10 ul dispensed into the well with iCMs and plates transferred to 37°C. Media was changed 24 hr after transfection. 8 independent wells of iCMs were transfected per construct to account for variability in plating and transfection efficiencies. A mammalian expression vector, pEF-GFP (Addgene, Plasmid 11154), was used to visually monitor transfection efficiency. At least 65–70% of the population of iCMs expressed GFP 24 hr hours post-transfection.

Luciferase activity was measured 24 hrhr after transfections using the Dual-Luciferase Reporter Assay System (Promega, Catalog#:E1980). After aspirating media, cells were rinsed with PBS once, and lysed with 20 uL of 1X Passive lysis buffer in the Luciferase assay kit. 15 min minutes after gentle shaking on an orbital shaker and complete lysis, the plate was stored at -80°C until further processing. Samples were prepared and luminescence measured according to Manufacturer's Assay protocol for 96-well plates using the Varioskan Flash Multimode Reader (Thermo Scientific).

## Data analysis

For all transfection wells, luminescence values of a blank non-transfection control were subtracted from all measured activity values. Firefly luciferase activity was then normalized to Renilla luciferase activity to control for transfection efficiency in each well. As luciferase reporter assay reagents decrease in activity during regular storage, the reference and alternate alleles of each reporter construct were spotted on the same 96-well plates to control for plate-to-plate variability in reagent activity. For each enhancer, we merged readings from multiple days by normalizing the activity of reporters to the reference allele. Each reporter construct was transfected into wells of at least two separate 96-well plates and readings for all wells were merged. Wells where Renilla luciferase activity (transfection control) was substantially lower (>90%) than neighboring wells were excluded from analyses. Statistical significance was determined by unpaired Student's t-test assuming equal variance. Minimum sample size of n=8 per enhancer construct was chosen to achieve 95% power for effect size (Cohen's d) of 2 (0.2 difference in activity between haplotypes with standard deviation of 0.1 normalized luciferase activity units) at p=0.05.

## Zebrafish antisense morpholino oligonucleotide-mediated knockdown and optical voltage mapping

Zebrafish (TuAB strain) were cared for according to standard techniques. All animal experiments were approved by the Partners Subcommittee on Research Animal Care (SRAC) and were conducted in compliance with the regulations published in the US National Institute of Health *Guide for the Care and Use of Laboratory Animals.* At the single cell stage, fertilized oocytes were injected with standardized concentrations and volumes of antisense morpholino oligonucleotides (5'CAATAGATGGCGCTGTGTACCTGTC3' and 5'AGAGCAGCCTGAAAGACAATAAAGA3' for *bves*, 5'GGTTAATCCACTCACCTGCCTGAAA3' and 5'CCGTCACTCGTATCCTGTTTTAGTG3' for *popdc3*, 5'3' and 'AGAAGTGTTTGCTCAGGTCACCTGT3' for *prep*, 5'GTTCAATTGTTTCTCACCTGCCAGA3' and 5'CTAATCCTGTGAAAGCAGAAGATCC3' for *popdc2*) dissolved in Danieau's solution (58 mM NaCl, 0.7mM KCl, 0.4 mM MgSO4, 0.6 mM Ca(NO3)2, 5.0 mM HEPES pH 7.6). Controls were injected with an equivalent dose of non-targeting morpholino of equal length but differing nucleotide composition (5'ATCCTCTTGAGGCGAACAAAGAGTC3'). RNA was harvested at 72 hr using TRIzol (Life Technologies) according to the manufacturer's instructions, cDNA synthesized by iScript reverse transcriptase (Bio-Rad Laboratories, Hercules, CA, Catalog #1708840) and semi-quantitative PCR was used to assess relative percentage of gene knockdown. All studies of morpholino efficacy are a result of samples obtained from three independent injections. For evaluation of ventricular action potential duration, embryo hearts were microdissected at 72 hr hours of development and stained with di-8-ANEPPS (Invitrogen, Catalog #). Cardiac contraction was arrested with 15 uM blebbistatin (Sigma-Aldrich). Hearts were then field paced at 2Hz and imaged at 1000 frames per second. Analysis of action potential durations was performed using an in-house developed MatLab program. The action potential duration at 80% repolarization was utilized for all analyses. A minimum *n* of 9 embryos was required for all ventricular action potential studies, based on power calculations for effect size (Cohen's d) of 1.5 at p=0.05. No animals were excluded from analyses unless ventricular depolarization could not be induced at 120 paces per minute. No randomization of samples or

blinding of investigators was utilized during these protocols. Statistical comparisons were performed using one-way ANOVA with Fisher's Least Significant Difference testing with all comparisons being to clutchmate controls. All distributions were normal, and variances between control and experimental groups were not statistically significant.

## Acknowledgements

We are thankful to E Gjoneska for advice on mammalian luciferase assays, P. Kheradpour for advice on TF motif analysis, G Quon and J. Wamstad for advice on epigenomics analyses and D Altshuler, L. Ward and V Agarwala for advice on human genetics analyses. We are grateful to the members of the Boyer and Kellis laboratories for helpful suggestions and discussions.

## Additional information

### Funding

| Funder | Grant reference number | Author |
|---|---|---|
| Canadian Institutes of Health Research | Doctoral Foreign Research Award | Xinchen Wang |
| Fondation Leducq | 14CVD01 | Wouter de Laat Patrick T Ellinor |
| National Institutes of Health | R01HL109004 | David J Milan |
| National Institutes of Health | R01HG004037 | Manolis Kellis |
| National Institutes of Health | R01HG008155 | Manolis Kellis |
| National Institutes of Health | R01GM113708 | Manolis Kellis |
| National Institutes of Health | U41HG007000 | Manolis Kellis |
| National Institutes of Health | U01HL098179 | Laurie A Boyer |
| American Heart Association | 15GRNT25670044 | Laurie A Boyer |

The funders had no role in study design, data collection and interpretation, or the decision to submit the work for publication.

### Author contributions

XW, Designed the project, Performed human genetics and epigenomics analyses, Performed 4C-seq experiments and analysis, Constructed luciferase reporter constructs and performed human cardiomyocyte luciferase assays, Wrote the manuscript, Drafting or revising the article; NRT, PTE, DJM, Performed zebrafish morpholino experiments, Acquisition of data, Analysis and interpretation of data, Drafting or revising the article; GR, Performed 4C-seq experiments and analysis, Constructed luciferase reporter constructs and performed human cardiomyocyte luciferase assays, Acquisition of data, Analysis and interpretation of data, Drafting or revising the article; RM, X-XN, JY, JL-M, EVD, Performed zebrafish morpholino experiments, Acquisition of data, Analysis and interpretation of data; PHLK, EdW, WdL, Performed 4C-seq experiments and analysis, Acquisition of data, Analysis and interpretation of data, Drafting or revising the article; VS, Constructed luciferase reporter constructs and performed human cardiomyocyte luciferase assays, Acquisition of data, Analysis and interpretation of data, Drafting or revising the article; EB, Constructed luciferase reporter constructs and performed human cardiomyocyte luciferase assays, Acquisition of data, Analysis and interpretation of data; PvdH, Acquisition of data, Contributed unpublished essential data or reagents; CN-C, Provided association results and analyses for sub-threshold QT interval loci, Analysis and interpretation of data, Drafting or revising the article, Contributed unpublished essential data or reagents; MK, Designed the project, Wrote the manuscript, Analysis and interpretation of data; LAB, Designed the project, Wrote the manuscript

### Author ORCIDs

Xinchen Wang, http://orcid.org/0000-0002-5198-6581

### Ethics

Animal experimentation: Zebrafish (TuAB strain) were cared for according to standard techniques. All animal experiments were approved by the Partners Subcommittee on Research Animal Care (SRAC, protocol #2005N000025) and were conducted in compliance with the regulations published in the US National Institute of Health Guide for the Care and Use of Laboratory Animals.

## Additional files

### Supplementary files

• Supplementary file 1. List of 112 QT/QRS loci compiled from NHGRI GWAS catalog, with corresponding study (sheet 1). List of 112 QT/QRS loci overlapping enhancers from each of four cardiac samples (sheet 2).

• Supplementary file 2. List of sub-threshold SNPs overlapping H3K4me1-defined LV enhancers (left columns) and H3K4me1/H3K27ac-defined active LV enhancers (right columns). Coordinates correspond to hg19 (sheet 1). List of candidate genes predicted to interact with sub-threshold loci within active LV enhancers by enhancer-gene linking algorithm (sheet 2). The list of sub-threshold SNPs is shorter than in Sheet #1 because some sub-threshold SNPs were excluded if the p-value was weaker than the most statistically significant SNP in LD ($r^2>0.2$, see Materials and methods).

• Supplementary file 3. List of primer sequences and enhancer sequences used for generating enhancer reporter constructs (sheet 1) and list of primers used for 4C-seq library construction (sheet 2).

### Major datasets

The following dataset was generated:

| Author(s) | Year | Dataset title | Dataset URL | Database, license, and accessibility information |
|---|---|---|---|---|
| Wang X | 2014 | Enhancers implicate new genes in QT interval below GWAS detection threshold | http://www.dtd.nlm.nih.gov/geo/query/acc.cgi?acc=GSE53567 | Publicly available at the NCBI Gene Expression Omnibus (accession no: GSE53567). |

The following previously published dataset was used:

| Author(s) | Year | Dataset title | Dataset URL | Database, license, and accessibility information |
|---|---|---|---|---|
| Roadmap Epigenomics Consortium | 2015 | Enhancer maps, RNA-seq, CpG methylation datasets | http://egg2.wustl.edu/roadmap/web_portal/ | Publicly available. Release 9 |

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
