## [Decision Letter]

Thank you for submitting your work entitled "Discovery and validation of sub-threshold GWAS loci using epigenomic signatures" for consideration by *eLife*. Your article has been favorably evaluated by Stylianos Antonarakis (Senior editor) and three reviewers, one of whom is a member of our Board of Reviewing Editors.

The reviewers have discussed the reviews with one another and the Reviewing editor has drafted this decision to help you prepare a revised submission.

Summary:

This paper presents a series of analyses that demonstrate clear evidence for enrichment of functionally-relevant genetic variants among sub genome-wide significant (GWS) loci from studies of heart parameters, and the value of using epigenetic signatures to further identify subsets that are likely to be real. The paper identifies one locus in particular and presents experimental data to support a hypothesis for a particular variant-gene interaction. Zebra-fish knock downs are also used to further support the involvement of a particular class of genes in QT interval / QRS duration. The authors suggest that the set of approaches they use have wide applicability. In the last couple of years several authors have demonstrated the value of epigenetic signatures as a way of homing in on signals in GWAS – or decomposing heritability. However, this is the first paper that we are aware of to follow up such work with experimental validation of sub-GWS signals.

There is a very large amount of work presented here and on the whole it is well-designed, logical and plausible. However, all three reviewers felt that there were substantial gaps in each of the pieces of work presented, which would have to be addressed in a revision. The key things that need to be revised are:

1) There has to be a quantitative attempt to estimate the FDR associated with the strategy of epigenetic prioritisation. It is very important for experimental researchers to know that a locus is 'real' before follow-up work starts – this is precisely why the GWS significance threshold is so stringent. An augmented set of experimental data along the lines presented here would enable this to be estimated.

2) In the luciferase and 4C experiments, only loci from the likely set are considered. In order to address the FDR issue, the same would have to be done for loci drawn from the no epigenetic signal set of loci with the same p value distribution. Such experiments would directly address the FDR issue.

3) In the analysis of matched, control SNPs, gene density needs to be accounted for.

4) All analyses should be carried out excluding GWS loci. In a paper about sub-GWS loci, including these is problematic.

5) In the fine-mapping example, we need to understand the prioritisation of rs1743292 better. There are other SNPs in LD with this in regulatory regions. Why were these not considered? We need to see a fine-mapping analysis of the signal with imputation (which is valid even though the region is not GWS) prior to focusing on the particular SNP.

6) In the morpholino experiments, there is apparently a lot of noise – such that the differences between those loci classed as positive and negative look relatively small. Additional replicates are needed to be convincing here.

7) There has to be a more up-to-date discussion of the relevant literature around biological signal lying in sub-GWS loci.

We appreciate that a lot is being asked for here and you may feel that such extensive revision is not feasible. One possibility is to remove some elements of the paper – e.g. the morpholino work – and focus on presenting a more complete story around the FDR part.

---

## [Author Response]

*There is a very large amount of work presented here and on the whole it is well-designed, logical and plausible. However, all three reviewers felt that there were substantial gaps in each of the pieces of work presented, which would have to be addressed in a revision. The key things that need to be revised are: 1) There has to be a quantitative attempt to estimate the FDR associated with the strategy of epigenetic prioritisation. It is very important for experimental researchers to know that a locus is 'real' before follow-up work starts – this is precisely why the GWS significance threshold is so stringent. An augmented set of experimental data along the lines presented here would enable this to be estimated.*

We thank the reviewers for this helpful comment. We agree that it is important to demonstrate that sub-threshold loci prioritized by epigenetic signals have a sufficiently high “true positive” rate to warrant follow-up investigations. In accordance with the reviewers’ request, we assayed the allele-specific differences in activity for an additional 11 enhancers that include 8 new sub-threshold loci.

In total, we now test 22 enhancers from 18 sub-threshold loci, observing that 13 loci (72%) display allele- specific activity in reporter assays. We quantitatively estimate that between 51.1%-89.8% (95% Bayesian confidence interval) of prioritized sub-threshold loci show allele-specific activity on transcription. These results suggest that the majority of sub-threshold loci prioritized by epigenomic annotations represent attractive candidates for follow-up by experimental researchers (Figure 4). We reproduce the text from the main paper:

“In total, we cloned allele-specific enhancer fragments from 22 cardiac enhancers that overlap SNPs from 18 independent sub-threshold loci, and performed quantitative luciferase assays in human iPSC-derived cardiomyocytes to determine whether the sub-threshold SNP genotypes influence enhancer activity (Methods). We observed that 13 of 18 sub-threshold loci (72.2%) contain an enhancer that drives luciferase activity in an allele-specific manner (Figure 4, Figure 4—figure supplement 1). Moreover, we estimate that between 51.1%-89.8% (95% Bayesian confidence interval) of prioritized sub-threshold loci show allele-specific activity on transcription, suggesting that the majority of sub-threshold loci identified by epigenomic prioritization do in fact have an impact on transcriptional enhancer activity.”

We also address the FDR concern by combining independent lines of evidence indicating that sub-threshold loci prioritized by epigenomics are significantly more attractive candidates for further study than those lack a relevant epigenetic signature.

*2) In the luciferase and 4C experiments, only loci from the likely set are considered. In order to address the FDR issue, the same would have to be done for loci drawn from the no epigenetic signal set of loci with the same p value distribution. Such experiments would directly address the FDR issue.* The reviewers raise an important point regarding the importance of demonstrating that sub-threshold loci prioritized by epigenomics have greater biological plausibility than those with no epigenetic signal.

We followed the reviewers’ recommendations and tested five sub-threshold loci marked by little or no epigenetic signal in human heart tissue using luciferase reporter assays (it was not feasible for us to perform additional 4C-seq experiments in the time frame). Four of these five loci show no allelic difference in activity, while surprisingly, two haplotypes at the fifth sub-threshold locus (rs9504919) differ in enhancer activity by 15% (p=0.0047, Figure 4—figure supplement 2).

We hypothesize that the allelic difference in activity at rs9504919 can be explained by the presence of an undiscovered cardiac enhancer. Specifically, the individuals contributing heart tissue to the Roadmap Epigenomics project are likely homozygous for the less active “C” allele at this locus (within the 1000 Genomes Project, 100% of 503 EUR individuals are homozygous C/C, and across all human populations, the C allele is present at a 95% frequency). Thus, the enhancer may only be detectable by epigenomic profiling studies if an individual with the correct genotype (A/C or A/A) is recruited. This touches upon a broader limitation of current epigenomic profiling studies that profile single “reference” individuals – these studies are unable to discover epigenomic signatures at sites where the individual carries a less active genotype, suggesting that more regulatory elements remain to be discovered by the profiling of the same tissues from additional individuals.

However, we respectfully wish to mention that the suggested experiment of testing the allele-specific activity of “negative control” sub-threshold loci can address the FDR issue. A negative result in this experiment (i.e. no allelic difference in activity) does not mean that these unmarked sub-threshold loci are not biologically relevant, but only that if they are “real”, they do not act through enhancer activity.

To better address the spirit of the reviewers’ comment on the effectiveness of using epigenetic signals to discriminate real sub-threshold loci from noise, we have re-written the text to emphasize two lines of evidence that support this point (subsection “Epigenomic prioritization discriminates biologically relevant sub-threshold loci”). We show that while all sub-threshold loci were identified using the same p-value cut-off of p<1x10_-4_, the loci marked by epigenetic signatures have significantly stronger GWAS signals in both the QT interval GWAS and a related GWAS for QRS duration (Figure 5).

Furthermore, we demonstrate that genetic perturbation of genes nearby the prioritized sub-threshold loci is significantly more likely to lead to cardiac conduction or contractility abnormalities compared to “negative control” sub-threshold loci outside cardiac enhancer elements (Figure 5). Notably, the tests in Figure 5 do not rely on assuming that sub-threshold loci affect enhancer activity, and thus the results demonstrate that epigenetic marks can prioritize biologically relevant sub-threshold loci.

*3) In the analysis of matched, control SNPs, gene density needs to be accounted for.* We thank the reviewers for this suggestion. We now include gene density (within a surrounding +/-500kb window) as a feature to control for in sampling the background distribution of control SNPs. We observe that the same patterns of greater enrichment of QT/QRS GWAS loci overlapping cardiac enhancers persist after account for gene density (Figure 7 adapted from Figure 1, but consistent patterns also observed everywhere else control SNPs were sampled for a background distribution).

Author response image 1.**DOI:**
http://dx.doi.org/10.7554/eLife.10557.024

*4) All analyses should be carried out excluding GWS loci. In a paper about sub-GWS loci, including these is problematic.* We apologize for the confusion regarding this important point. We started our study by identifying 112 QT/QRS GWAS loci from GWASs published up to July 2013 and used this set to discover the common characteristics of GWAS loci outlined in Figure 1 and Figure 2 (e.g. cardiac enhancer overlap and the greater density of specific epigenetic marks). The GWS loci in Figure 3 that remain after removal of the 112 QT/QRS GWAS loci are loci that only reach genome-wide significance in the larger 2014 Arking et al.GWAS. These newly GWS loci were purposefully held out from the initial set of 112 QT/QRS loci so that they can act as an independent validation set for testing the effectiveness of epigenomic prioritization, as these newly GWS loci are almost certainly biologically real.

We have clarified this point in the text at multiple places:

”We also collected GWAS loci from a later meta-analysis of QT interval studies, published in June 2014 by Arking et al., which we held out from the aforementioned 112 QT/QRS loci as a validation dataset for subsequent analyses.”

“To test this idea, we used SNP summary statistics from the Arking et al.(2014) QT interval GWAS study we had earlier held out as a validation dataset (Arking et al., 2014). These summary statistics include the 112 QT/QRS loci identified by prior GWASs (red dots, bottom, Figure 3), as well as loci that reach genome-wide significance in the larger meta-analysis cohort but were not discovered in any previous GWAS (and therefore were not included in the 112 QT/QRS loci used for enrichment analyses above, gold dots, bottom, Figure 3).”

However, to address the reviewers’ concern regarding the additional GWS loci, we have performed the same enrichment analysis after also excluding the held-out GWS and any SNPs within +/- 1Mb. The plot is included in Figure 8, and we observe the same enrichment pattern as in Figure 3.

Author response image 2.**DOI:**
http://dx.doi.org/10.7554/eLife.10557.025

*5) In the fine-mapping example, we need to understand the prioritisation of rs1743292 better. There are other SNPs in LD with this in regulatory regions. Why were these not considered? We need to see a fine-mapping analysis of the signal with imputation (which is valid even though the region is not GWS) prior to focusing on the particular SNP.* We thank the editors and reviewers for this suggestion. Unfortunately, we do not have access to the individual-level genotype information to perform statistical fine-mapping analysis. However, to address the reviewers’ concern, in the revised manuscript we now consider all putative causal SNPs in the rs1743292 locus as determined by the combination of GWAS signal and epigenetic signal. As described in the subsection “Sub-threshold locus at rs1743292/rs1772203 functionally disrupts enhancer activity”, the rs1743292 locus contains 15 SNPs (r_2_≥0.8). 8 of 15 have a p-value reported by Arking et al.of less than 1x10^-^_4_ and an additional 2 do not have reported p-values. We considered these 10 SNPs as possible causal SNPs in the locus, and in our revised manuscript we study the 3 SNPs out of 10 that overlap active LV enhancers (rs1743292, rs112332323, rs1772203).

As also described in the aforementioned subsection, both the rs1743292/rs112332323 enhancer and the rs1772203 enhancer show allele-specific enhancer activity, and both enhancers form detectable 3D interactions with the *BVES* and *POPDC3* promoters (Figure 6, Gg).

Thus, we now believe that genetic variation in the rs1743292 locus affects both enhancers, and that rs1772203 is an additional potential causal SNP that contributes to the activity of this locus.

*6) In the morpholino experiments, there is apparently a lot of noise – such that the differences between those loci classed as positive and negative look relatively small. Additional replicates are needed to be convincing here.* We thank the reviewers for the suggestion. It was not feasible for us to perform additional morpholino experiments in the time frame, so we followed the editors’ suggestion below and moved most of the zebrafish section from the main text to a supplemental figure (Figure 6—figure supplement 2). In the revised manuscript, we now focus primarily on the organism-level phenotype validation of the rs1743292 locus using mouse phenotypes, showing that loss of *Bves*, but not *Popdc3* or *Prep*, leads to cardiac contractility and conduction defects (Figure 6). However, we still present the zebrafish optical voltage mapping results in the supplemental figure because we believe the phenotypic agreement between *bves* knockdown in zebrafish and *Bves* knockout in mouse strengthens our case for *BVES* as the causal gene in the locus.

*7) There has to be a more up-to-date discussion of the relevant literature around biological signal lying in sub-GWS loci.* We thank the reviewers for this suggestion. In the revised manuscript we now include an up-to-date discussion of the literature around sub-GWS loci. Briefly, three recent studies (PMIDs: 22955828, 26414678, 25439723) have observed the general enrichment of either sub-threshold SNPs or SNPs that explain a disproportionately high amount of heritability in cell type-specific regulatory elements, however to our knowledge no study has moved beyond this observation to investigate any individual sub- threshold locus or tracked down the relevant genes or underlying pathways. This latter point is critical, as the editors and reviewers mention in point #1: it is “very important for experimental researchers to know that a locus is 'real' before follow-up work starts”.

We have added the following to the Discussion: “Studies of genetic heritability have indicated that many additional loci lie below the genome-wide significance threshold (Yang et al., 2011). […] The work presented represents a first step towards deciphering this signal and opens the door for the discovery of greater numbers of disease loci, genes, and pathways.”